# PRIVATE IMAGE RECONSTRUCTION FROM SYSTEM SIDE CHANNELS USING GENERATIVE MODELS

**Yuanyuan Yuan & Shuai Wang**[*]
Department of Computer Science and Engineering
The Hong Kong University of Science and Technology
Clear Water bay, Hong Kong SAR
{yyuanaq,shuaiw}@cse.ust.hk

**Junping Zhang**
Department of Computer Science
Fudan University
Shanghai, China
jpzhang@fudan.edu.cn

## ABSTRACT

System side channels denote effects imposed on the underlying system and hardware when running a program, such as its accessed CPU cache lines. Side channel analysis (SCA) allows attackers to infer program secrets based on observed side channel logs. Given the ever-growing adoption of machine learning as a service (MLaaS), image analysis software on cloud platforms has been exploited by reconstructing private user images from system side channels. Nevertheless, to date, SCA is still highly challenging, requiring technical knowledge of victim software's internal operations. For existing SCA attacks, comprehending such internal operations requires heavyweight program analysis or manual efforts.

This research proposes an attack framework to reconstruct private user images processed by media software via system side channels. The framework forms an effective workflow by incorporating convolutional networks, variational autoencoders, and generative adversarial networks. Our evaluation of two popular side channels shows that the reconstructed images consistently match user inputs, making privacy leakage attacks more practical. We also show surprising results that even one-bit data read/write pattern side channels, which are deemed minimally informative, can be used to reconstruct quality images using our framework.

## 1 INTRODUCTION

Side channel analysis (SCA) recovers program secrets based on the victim program's nonfunctional characteristics (e.g., its execution time) that depend on the values of program secrets. SCA constitutes a major threat in today's system and hardware security landscape. System side channels, such as CPU cache accesses and operating system (OS) page table accesses made by the victim software, are widely used to recover program secrets under various real-world scenarios (Gullasch et al., 2011; Aciicmez & Koc, 2006; Wu et al., 2012; Hähnel et al., 2017; Xu et al., 2015; Yarom et al., 2017).

To conduct SCA, attackers first conduct an online phase to log a trace of side channel data points made by the victim software (e.g., its accessed CPU cache lines). Then, attackers launch an offline phase to analyze the logged trace and infer secrets (e.g., private inputs). Enabled by advances in system research, the online phase can be performed smoothly (Xu et al., 2015). Nevertheless, the offline phase is challenging, requiring comprehension of victim software's input-relevant operations and how such operations influence side channels. The influence is program-specific and obscure (see an example in Fig. 1). Even worse, side channel data points made by real-world software are usually highly noisy. For instance, executing `libjpeg` (libjpeg, 2020) to decompress one unknown JPEG image produces a trace of over 700K side channel data points, where only a small portion depends on the image content. Identifying such input-dependent data points from over 700K records is extremely difficult.

Launching SCA to recover images processed by media software constitutes a common threat in the era of cloud computing (Xu et al., 2015; Hähnel et al., 2017), especially when machine learning as a service (MLaaS) is substantially offered (e.g., for face recognition). When envisioning the high

---

[*]Corresponding Author

risk of violating user privacy, there is a demanding need to understand the adversarial capability of reconstructing private images with SCA. To date, the offline inference phase of existing SCA attacks requires lots of manual efforts with heuristics (Xu et al., 2015; Hähnel et al., 2017). While some preliminary studies explore to use AI models to infer secrets (Hospodar et al., 2011; Kim et al., 2019; Cagli et al., 2017; Hettwer et al., 2018), their approaches are primarily driven by *classification*, i.e., predicting whether a particular bit of crypto key is 0 or 1. In contrast, reconstructing user private images requires to synthesize and enhance images from a more holistic perspective.

Recent advances in generative models, such as generative adversarial network (GAN) and variational autoencoder (VAE), have enabled a major thrust in image reconstruction, given subtle signals in even cross-modal settings, e.g., voice-to-face or text-to-image (Radford et al., 2016; Reed et al., 2016; Wen et al., 2019; Hong et al., 2018b). Inspired by this breakthrough, we propose an SCA framework using generative models. Given a trace of side channel data points made by image analysis software (e.g., `libjpeg`) when processing a user input, we reconstruct an image visually similar to the input. Each logged side channel trace, containing around a million records, is first encoded into a matrix and pre-processed by a convolutional neural network (CNN) for feature extraction. Then, a VAE network with a learned prior (referred to as VAE-LP) is employed to reconstruct an image with a holistic visual appearance. We further supplement VAE-LP with a GAN model to enhance the recovered image with vivid details. The GAN generator yields the final output.

Our attack exploits media libraries, `libjpeg` (libjpeg, 2020) and `uPNG` (Middleditch, 2010), using two popular side channels, CPU cache line accesses and OS page table accesses. Our attack is independent of the underlying computing infrastructure (i.e., OS, hardware, image library implementation). We require enough side channel logs for training, which is consistently assumed by previous works (Heuser & Zohner, 2012; Maghrebi et al., 2016). While existing attacks particularly target `libjpeg` and leverage domain knowledge, system hacking, and manual efforts to infer pixel values (Xu et al., 2015; Hähnel et al., 2017), we show that images with many details can be reconstructed in an end-to-end manner. We also show surprising results that enabled by our framework, side channel traces composing *one-bit data read/write* patterns, which prima facie seems minimally informative, suffice recovering images. We conduct qualitative and quantitative evaluations on specific and general datasets representing daily images that can violate privacy if leaked. The recovered images manifest consistent visual appearances with private inputs. The recovered images also exhibit high discriminability: each recovered image (e.g., a face) can be matched to its reference input among many candidates with high accuracy. In summary, we make the following contributions:

- At the conceptual level, we present the first generative model-based SCA. Our novel approach learns how program inputs influence system side channels from historical side channel logs to reconstruct user private images automatically. We, for the first time, demonstrate surprisingly effective attacks toward even low-resolution side channels like one-bit data read/write access patterns.

- At the technical level, we design an effective framework by incorporating various design principles to facilitate image reconstruction from side channels. Our framework pipelines 2D CNN, VAE-LP, and GAN models to systematically enhance the quality of generated images.

- At the empirical level, our evaluations show that the proposed framework can generate images with vivid details and are closely similar to reference inputs. The reconstructed images show high discriminability, making privacy leakage attacks more practical.

This is the first paper to conduct SCA with generative models, revealing new SCA opportunities and unknown threats. Our code is at `https://github.com/genSCA/genSCA`.

## 2 BACKGROUND

To formulate SCA, let the attacked program be $P$ and its input domain be $I$. For a deterministic and terminating program $P$, the program execution can be modeled as a mapping $P : I \rightarrow E$ where $E$ represents program runtime behavior (e.g., memory access). As a common assumption (Hähnel et al., 2017), program inputs are *private* and *profitable* for attackers. Since different inputs $i, i' \in I$ can likely induce different $e, e' \in E$, using input-dependent $e \in E$ enables to infer $i$.

Modern computer architectures have primarily zeroed the possibility for adversaries to log $e \in E$. Nevertheless, an attacker's view on $P$ can be modeled as a function $view : E \rightarrow O$ that maps $E$ to *side channel* observations $O$. Hence, the composition $(view \circ P) : I \rightarrow O$ maps inputs to side channel data points that can be logged by attackers. The $view$ indicates the attacker's capability, and for typical system security scenarios, the $view$ is formulated as $view : E^{mem} \rightarrow O_{cache} \cup$

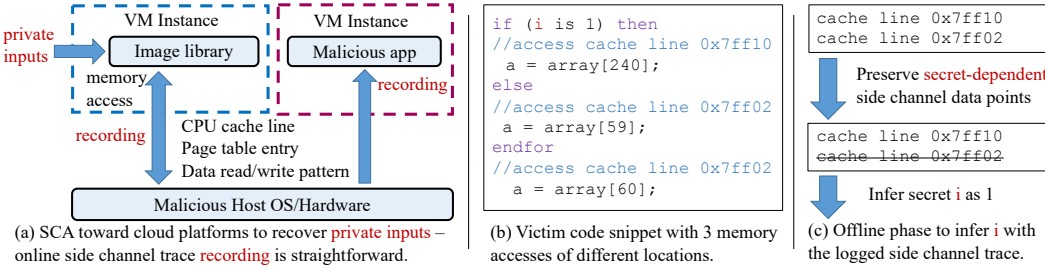

Figure 1: Performing SCA to exploit cloud computing platforms.

$O_{page}$, where $E^{mem}$ denotes a trace of accessed memory locations when executing $P$ with $i$, and $O_{cache}$ and $O_{page}$ represent CPU cache and OS page table side channels, respectively. Despite being unable to monitor $E^{mem}$, attackers can log accessed cache lines $O_{cache}$ or page table entries $O_{page}$ derived from $E^{mem}$. Attackers then infer $E^{mem}$ and recover $i$. We now concretize the procedure by introducing how SCA is used to exploit cloud platforms in a two-step approach as follows:

**Online Phase to Record $O$.** Considering a cloud environment in Fig. 1(a), where two users, one normal and one malicious, deploy two virtual machine (VM) instances on the host. Private images $i \in I$ uploaded by users are processed by media library $P$ within the left VM. Modern computer design, e.g., Intel SGX (Intel, 2014), guarantees that $i \in I$ and the execution of $P$ cannot be viewed from outside the VM. However, when processing $i$, $P$ usually imposes a large volume of CPU cache and page table accesses, which, as shown in Fig. 1(a), can be recorded by the co-located malicious VM or the malicious host OS in a fully automated manner (Han et al., 2017; Chiang et al., 2015; Liu et al., 2015a; Xu et al., 2015; Hähnel et al., 2017).

**Offline Phase to Infer $i$.** Once side channel traces $o \in O$ are collected, an offline phase is conducted to infer $(view \circ P)^{-1} : O \to I$ and recover $i$. Fig. 1(b) presents a sample code, where depending on values of input $i$, different memory locations (and cache lines) will be visited. Fig. 1(c) shows the corresponding trace of logged cache side channel records. To infer $i$, attackers eliminate the second record (since it is input-independent), and infer $i$ as 1 according to the first record.

Attackers anticipate to 1) pinpointing a subset of records $o^* \subseteq o$ that depend on $i$, and to 2) recovering the mapping from $o^*$ to $i$. However, real-world side channel traces (e.g., generated by uPNG) could contain over one million records, where only a tiny portion $o^*$ is *input-dependent*. Even worse, constructing the mapping between $i$ and $o^*$ requires a deep understanding of program control flows (e.g., how i affects program execution and induces cache accesses in Fig. 1(b)). To date, these tasks require either manual effort (Xu et al., 2015; Hähnel et al., 2017) or formal analysis (Doychev et al., 2013; Wang et al., 2017; 2019), which are program-specific and error-prone with low scalability.

Existing research tackles the offline phase challenge by proposing *profiling-based SCA* (Maghrebi et al., 2016; Hettwer et al., 2018; Kim et al., 2019), where models are trained to approximate $(view \circ P)^{-1} : O \to I$. However, existing work focuses on predicting particular bits of crypto keys from succinct side channel traces, e.g., a few hundred records (Hettwer et al., 2020). In contrast, this is the first work shows that by incorporating generative models, SCA can be conducted to exploit real-world media libraries and holistically reconstruct high-quality and discriminable images.

## 3   THE PROPOSED FRAMEWORK

A common assumption shared by SCA (Heuser & Zohner, 2012; Hähnel et al., 2017; Xu et al., 2015) is that the attackers can profile the victim software locally or remotely with training inputs and collect corresponding side channel traces. We train a model to learn how different inputs can influence side channel traces. Then, given a side channel trace logged when processing an unknown image, our framework reconstructs an image that is visually similar to the unknown input.

Our framework has two pipelined modules (see Fig. 2). Given a side channel trace $T_i$ corresponding to processing an image $i$, we first encode $T_i$ into a matrix. The encoded matrix will be fed to the VAE-LP module to generate image $\hat{i}_{trace}$, and we further use GAN to denoise $\hat{i}_{trace}$ and yield the final output $\hat{i}_{GAN}$. We now elaborate on each module. More details are given in Appendix B.

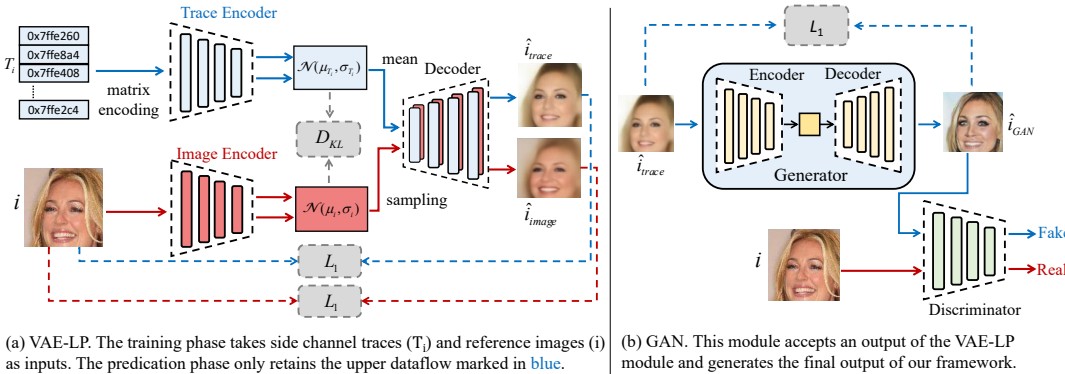

(a) VAE-LP. The training phase takes side channel traces ($T_i$) and reference images ($i$) as inputs. The predication phase only retains the upper dataflow marked in blue.

(b) GAN. This module accepts an output of the VAE-LP module and generates the final output of our framework.

Figure 2: The proposed generative framework for recovering private images from side channel traces.

## 3.1 SIDE CHANNEL TRACE ENCODING

Real-world software is highly complex, and processing one image could generate a huge amount of records, where only a few records are *secret-dependent*. Processing overlong traces for previous attacks is very difficult and requires considerable domain-specific knowledge, expertise, and even manual efforts to locate and remove irrelevant records (Xu et al., 2015; Hähnel et al., 2017).

Despite the general difficulty of processing overlong traces (each trace contains about 700K to 1.3M data points in our evaluation), we note that adjacent records on a side channel trace are often derived from the same or related modules (e.g., functions) of the victim software. Hence, we "fold" each side channel trace into a $N \times N \times K$ matrix to approximate spatial locality which can be further exploited by CNNs. A trace is first divided into $K$ segments, where $N$ adjacent points in a segment are put into one row, in total $N$ rows. We do zero padding. CNNs are deployed in the trace encoder of VAE-LP to process the encoded matrices. Overall, we have no assumption of the access pattern or the convolutional structure of the inputs. Side channel traces are generally sparse, where only a small portion is private-related (see Appendix J for experimental information). To smoothly process the side channel traces with generative models, we thus employ CNN models to pre-process side channel traces.

## 3.2 THE VAE-LP MODULE

VAE-LP extends standard VAE by replacing its fixed Gaussian prior with a *learned prior* (Denton & Fergus, 2018), which represents the latent distribution of side channel traces. VAE-LP is trained using both real-life images and their corresponding side channel traces. By incorporating the side channel trace encoder, we can extract latent representations from the logged side channel data points. Simultaneously, by integrating corresponding reference images during training, we provide a guideline to help the image decoder to generate quality images. As shown in Fig. 2(a), the trace encoder $Enc_{trace}$ (marked in blue) employs 2D CNNs and can extract features from side channel traces $T_i$ in the encoded matrices. The output of $Enc_{trace}$ constitutes the learned prior distribution of latent variable, namely $p(z_{T_i})$, rather than a fixed Gaussian distribution that VAE usually is. The decoder $Dec$ takes the mean of $p(z_{T_i})$ as its input and outputs the image generated from side channel traces. The training phase also employs the image encoder $Enc_{image}$ (marked in red), which accepts reference images $i$ and outputs $q(z_i|i)$.

We train the VAE-LP network by performing forward propagation separately for two different data resources. We then use two generated images to compute reconstruction loss and perform one iteration of backward propagation. Let $\hat{i}_{trace}$ and $Dec_{trace}$ be the generated image and decoder $Dec$ by conducting forward propagation with only $T_i$. Similarly, let $\hat{i}_{image}$ and $Dec_{image}$ be the generated image and $Dec$ by conducting forward propagation with only $i$. Parameters of $Dec_{trace}$ and $Dec_{image}$ are shared in the training phase, and the loss of the VAE-LP module is defined as follows:

$$Loss_{VAE-LP} = L_1(i, \hat{i}_{image}) + L_1(i, \hat{i}_{trace}) + \beta D_{KL}(q(z_i|i)||p(z_{T_i}))$$

where three terms, namely, i) a reconstruction loss $L_1(i, \hat{i}_{image})$ derived from the reference input, ii) a reconstruction loss $L_1(i, \hat{i}_{trace})$ derived from the side channel trace, and iii) a KL-divergence

Table 1: Side channels derived from a memory access made by victim software using address $addr$.

| Side Channel Name | Side Channel Record Calculation |
|---|---|
| CPU Cache Line Index | $addr \gg L$ where $L$, denoting cache line size, is 6 on modern x86 architectures. |
| OS Page Table Index | $addr \,\&\, (\sim M)$ where $M$, denoting PAGE_MASK, is 4095 on modern x86 architectures. |
| Data Read/Write Pattern | one bit (0/1) denoting whether this memory access is a read or write operation. |

that forces $q(z_i|i)$ to be close to $p(z_{T_i})$ are subsumed. During the generation phase, we remove $Enc_{image}$ and $Dec_{trace}$. $Enc_{trace}$ and $Dec_{image}$ are retained to yield $\hat{i}_{trace}$ given a logged $T_i$. The generative process is given by:

$$z_{T_i} \sim p(z_{T_i}) \quad \hat{i}_{trace} \sim p(i|z_{T_i})$$

### 3.3 THE GAN MODULE

VAE-LP module is seen to recover $\hat{i}_{trace}$ with relatively coarse-grained information (see Sec. 5.1). As will be elaborated in Sec. 4, different private inputs can manifest *identical* side channel patterns. Hence, some details are inevitably missing during input reconstruction. To tackle this inherent limitations, we further deploy a GAN module (see Fig. 2(b)) which takes the output of the VAE-LP module, $\hat{i}_{trace}$, and generates the final output $\hat{i}_{GAN}$. To smoothly refine $\hat{i}_{trace}$, we employ an autoencoder as the generator $G$ of GAN. The loss of the extended GAN model is defined as follows:

$$Loss_{GAN} = \gamma L_1(G(\hat{i}_{trace}), \hat{i}_{trace}) + \mathbb{E}_{i \sim p(i)}[\log D(i)] + \mathbb{E}_{\hat{i}_{trace} \sim p(\hat{i}_{trace})}[\log(1 - D(G(\hat{i}_{trace})))]$$

Compared with standard GAN, we extend the loss function with $L_1$ loss of $\hat{i}_{trace}$ and $G(\hat{i}_{trace})$ with a weight of $\gamma$ to force $G$ to retain the holistic visual appearance delivered by $\hat{i}_{trace}$. $L_1$ loss is generally acknowledged to perform better on capturing the low-frequency part of an image (Isola et al., 2017). Indeed, our evaluation shows that $L_1$ loss, as a common setting, sufficiently conducts SCA and recovers user private inputs of high quality.

## 4 ATTACK SETUP

As introduced in Sec. 2, popular system side channels are primarily derived from program memory accesses. Let $addr$ be the address of a memory location accessed by the victim software, Table 1 reports three utilized side channels and how they are derived from memory accesses. Cache line and page table side channels are commonly used for exploitation (Hähnel et al., 2017; Yarom & Falkner, 2014). Furthermore, enabled by our framework, a very low-resolution side channel of data read/write access patterns can, for the first time, be used to reconstruct high-quality images. Fig. 1 holistically depicts how attackers can monitor cache and page table side channels. Data read/write patterns can be similarly recorded by monitoring how caches or page tables are visited.

Table 1 shows that different $addr$ can be mapped to the same side channel record. Similarly, different inputs can induce identical memory address $addr$. For instance, in Fig. 1(b) `array[59]` and `array[60]` will always be executed as long as $i \neq 1$. Two layers of many-to-one mapping amplify the uncertainties of synthesizing discriminable images of high quality. It is easy to see that we are **not** simply mapping a trace back to an image.

Each memory address $addr$ has 48 bits, denoting a large range of values. We *normalize* the memory address value (discrete integers) into continuous values within $[0, 1]$. Overall, while arbitrary 48-bit integers have a large range, side channel data points indeed vary within a small range. For instance, for cache based side channels, the possible values are limited by the total number of CPU cache units. In all, side channel data points are large values ranging in a relatively small range.

**Attack Target.** We attack two media libraries, `libjpeg` and `uPNG`, to reconstruct private user images of JPEG and PNG formats. Previous image reconstruction SCA (Xu et al., 2015; Hähnel et al., 2017) only exploit `libjpeg`. PNG and JPEG denote very popular image compression standards, and given an image in JPEG/PNG format, `libjpeg` and `uPNG` can reverse the compression to generate a bitmap image as the basis of many image analysis tools, e.g., the Python `Pillow` library (Clark, 2020). The decompression process introduces many input-dependent memory accesses which, from the attacker's perspective, can be reflected on side channels according to Table 1.

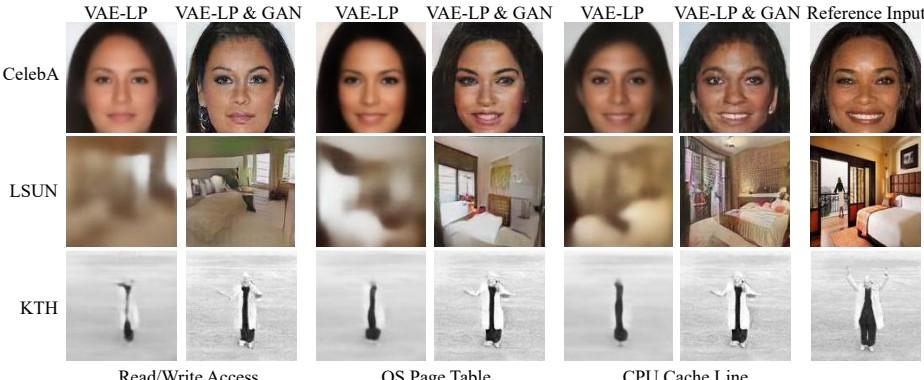

Figure 3: Qualitative evaluation results of three datasets in terms of three different side channels. Corresponding reference images (i.e., private user inputs) are presented on the rightmost column for comparison.

We share *common assumptions* with existing profiling-based SCA (Hospodar et al., 2011; Heuser & Zohner, 2012) that side channel traces have been well prepared for use. For our experiments, we use `Pin` (Luk et al., 2005), a runtime monitoring tool, to intercept memory accesses of victim software. A logged memory access trace will be converted into three separate side channel traces according to Table 1, denoting the attacker's view on $i$. Each side channel trace generated by `libjpeg` or `uPNG` contains 700K to 1.3M records. See Appendix A for attack setup details. We evaluate traces logged via different side channels *separately*. Evaluating the composition of side channels (i.e., a "mega-side channel") is not aligned with how real-world SCA is typically launched.

## 5 EVALUATION

We present the first systematic approach to reconstructing images from side channels. There is no previous research for empirical comparison. Two closely related works provide no tools for use (Xu et al., 2015; Hähnel et al., 2017). As disclosed in their papers, manual efforts are extensively used to reconstruct images. For instance, both methods treat image color recovery as a separate task, by iterating multiple reconstruction trials and manually picking one with relatively better visual effect. Xu et al. (2015) exploit page table side channels and colors are rarely recovered. Hähnel et al. (2017) recover adequate image colors but only exploit finer-grained cache side channels. Also, domain-specific knowledge on `libjpeg` is required to locate a tiny portion of secret-dependent side channel data points for use. In contrast, we present an end-to-end approach to recovering colorful images with high quality, by directly analyzing a page table or cache side channel trace of up to 1.3M records. Our attack treats victim software (`libjpeg` or `uPNG`) as a "black-box" (no need for source code) and is independent of any underlying computing infrastructure details.

∎ **Benchmarks.** Three datasets are primarily used in the evaluation, containing typical daily images that could violate privacy if leaked to adversaries. Consistent with existing research reconstructing images from audio recording (Wen et al., 2019; Oh et al., 2019), we accelerate the model training using images of $3 \times 128 \times 128$ pixels. Wen et al. (2019) use images of an even smaller size ($3 \times 64 \times 64$). See Appendix B for model implementation and training details.

**(i) Large-scale CelebFaces Attributes (CelebA)** (Liu et al., 2015b) contains about 200K celebrity face images. We randomly select 80K images for training and 20K images for testing.

**(ii) KTH Human Actions (KTH)** (Laptev & Lindeberg, 2004) contains videos of six actions made by 25 persons in 4 directions. For each action, we randomly select videos of 20 persons for training and use the rest for testing. We have 40K images for training and 10K images for testing.

**(iii) LSUN Bedroom Scene (LSUN)** (Yu et al., 2015) contains images of typical bedroom scenes. We randomly select 80K images for training and 20K images for testing.

### 5.1 QUALITATIVE EVALUATION RESULTS

Fig. 3 shows the reconstructed images in different settings. In addition to reporting the final outputs (i.e., the "VAE-LP & GAN" column), we also report images generated only from the VAE-LP module in the "VAE-LP" column for comparison. More results are given in Appendix C. For most of the cases, the reconstructed images and reference images show consistent visual appearances,

Table 2: Discriminability evaluation. We present success rates of random guess (RG) for the comparison.

| | $k = 1 \ N = 100$ | | | | $k = 5 \ N = 100$ | | | | $k = 20 \ N = 100$ | | | |
| | CelebA | KTH | LSUN | RG | CelebA | KTH | LSUN | RG | CelebA | KTH | LSUN | RG |
|---|---|---|---|---|---|---|---|---|---|---|---|---|
| CPU Cache Line | 47.28% | 38.22% | 10.00% | 1% | 74.98% | 63.56% | 28.74% | 5% | 90.70% | 83.72% | 58.88% | 20% |
| OS Page Table | 16.16% | 35.86% | 2.26% | 1% | 37.96% | 61.28% | 8.62% | 5% | 65.14% | 82.22% | 26.42% | 20% |
| Read/Write Pattern | 19.58% | 33.44% | 2.24% | 1% | 42.44% | 58.30% | 8.28% | 5% | 67.82% | 80.14% | 26.64% | 20% |
| | $k = 1 \ N = 500$ | | | | $k = 5 \ N = 500$ | | | | $k = 20 \ N = 500$ | | | |
| | CelebA | KTH | LSUN | RG | CelebA | KTH | LSUN | RG | CelebA | KTH | LSUN | RG |
| CPU Cache Line | 28.68% | 24.50% | 3.04% | 0.2% | 52.22% | 41.24% | 10.90% | 1% | 72.98% | 61.08% | 25.92% | 4% |
| OS Page Table | 6.56% | 22.58% | 0.46% | 0.2% | 17.90% | 38.80% | 2.42% | 1% | 35.22% | 58.90% | 7.30% | 4% |
| Read/Write Pattern | 8.42% | 20.64% | 0.50% | 0.2% | 21.18% | 36.22% | 2.54% | 1% | 39.84% | 55.74% | 7.08% | 4% |

such as gender, skin color, bedroom window, and human gesture. Images in the LSUN dataset contain many subtle bedroom details, imposing relatively higher challenge for reconstruction.

Realistic and recognizable images can be recovered using cache side channels (especially for LSUN) while images recovered from the other side channels are relatively blurry. As explained in Table 1, cache line indices ($addr \gg 6$) are closer to the memory address $addr$ (only missing the lowest 6 bits), while page table indices eliminate the lowest 12 bits from $addr$ (typically lower bits in $addr$ are informative and likely influenced by inputs), and each read/write pattern has only one bit.

Compared with images generated by VAE-LP, the GAN module enhances the image quality by adding details and sharpening the blurred regions. GAN may overly enhance the image quality (e.g., the first LSUN case with jungle green wallpaper). However, GAN is indeed vital to exploit user privacy. For example, considering the first human gesture case in KTH, where the image reconstructed from cache side channels contains a "black bar" when using VAE-LP. The GAN module enhances this obscure image and reconstructs the human gesture, thus violating user privacy.

## 5.2 QUANTITATIVE EVALUATION RESULTS

To assess the generated images w.r.t. discriminability, we first study the accuracy of matching a reconstructed image $\hat{i}$ to its reference input $i$. To do so, we form a set of $N$ images which include the reference input $i$ and $N - 1$ images randomly selected from our testing dataset. We then measure whether $i$ appears in the top-$k$ most similar images of $\hat{i}$. Conceptually, we mimic a de-anonymization attack of user identity, where $N$ scopes the search space attackers are facing (e.g., all subscribers of a cloud service). We use a perceptually-based similarity metric, SSIM (Wang et al., 2004), to quantify structural-level similarity between the reconstructed images and reference inputs.

Table 2 reports the evaluation results of six practical settings. Consistent with Fig. 3, cache side channels help to reconstruct $\hat{i}$ with better discriminability (highest accuracy for all settings in Table 2). LSUN images have lower accuracy. As shown in Fig. 3, images in LSUN contain many subtle bedroom details and deem challenging for discrimination. We achieve the highest accuracy when $k = 20$ and $N = 100$, while the accuracy, as expected, decreases in more challenging settings (e.g., when $k = 1$ and $N = 500$). Evaluations consistently outperform the baseline — *random guess*. For instance, while the accuracy of random guess when $k = 1$ and $N = 500$ is 0.2%, we achieve higher accuracy (range from 0.46% to 28.68%) across all settings. Appendix D also conducts this evaluation using images generated from only VAE-LP. We report that better discriminability can be achieved for all datasets when supplementing VAE-LP with GAN.

For face images in CelebA, we also study how well different facial attributes are being captured in the reconstructed images. We use Face++ (fac, 2020), a commercial image analysis service, to classify reconstructed images and reference inputs w.r.t. age and gender attributes. Fig. 4 reports the confusion matrices for age and gender attributes and distributions of training data for the reference. The reconstructed images are produced using cache side channels. Overall, we achieve a good agreement for both male and female labels. We also observe correlated classification results for most age groups. The age distribution indicates that early adulthood (20–40) and middle age (40–60) mostly dominate the dataset, which presumably induces biases in the age confusion matrix. Similarly, "male" has a smaller representation in the training set, potentially explaining its lower agreement in the confusion matrix. Appendix D also conducts this evaluation using only VAE-LP or using other side channels where comparable results can be achieved.

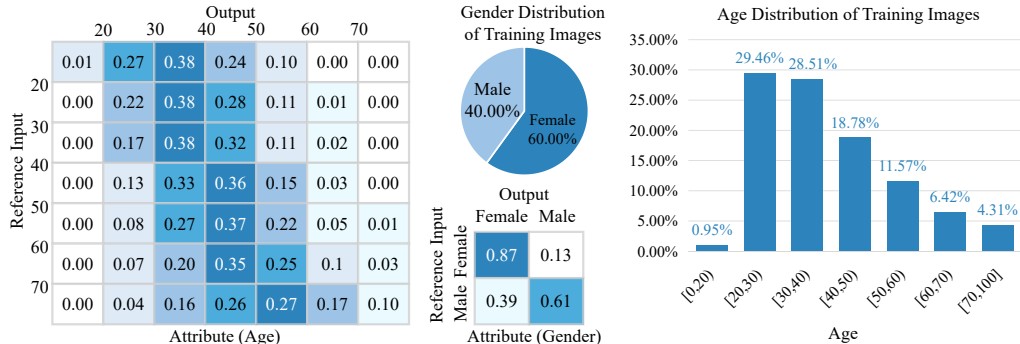

Figure 4: Facial attribute evaluation in terms of cache side channels. Confusion matrices (with row-wise normalization) compare the age and gender classification results on our recovered images and those obtained from reference inputs. We also report the age and gender distributions in the training set for the reference.

Table 3: Generalizability evaluation.

| $(k, N)$ | $(1, 100)$ | $(5, 100)$ | $(20, 100)$ | $(1, 500)$ | $(5, 500)$ | $(20, 500)$ |
|---|---|---|---|---|---|---|
| uPNG: SCA model trained with CelebA | 18.8% | 40.6% | 66.0% | 8.3% | 20.7% | 38.4% |
| libjpeg: SCA model trained with mini-Imagenet | 4.4% | 17.1% | 39.8% | 1.1% | 5.0% | 15.0% |
| Baseline — random guess | 1% | 5% | 20% | 0.2% | 1% | 4% |

## 5.3 GENERALIZABILITY

This section explores the generalizability of our SCA. We launch attacks toward uPNG to illustrate that our method is *independent* of specific software implementation or image formats. For uPNG experiments, we evaluate attacks on the CelebA dataset using cache side channels. As shown in table Table 3, our attack can recover discriminable images and largely outperforms the baseline (random guess) in terms of privacy inference. See Appendix E for the reconstructed images. We also benchmark our attack to synthesize arbitrary images without using *specific* types of images to train the model. We instead use a *general* training dataset, mini-Imagenet. We use cache side channels to exploit libjpeg. Table 3 illustrates that considerable privacy is leaked using mini-Imagenet as the training set, and for all settings, our SCA largely outperforms the baseline.

The recovered images when using mini-Imagenet as the training data are visually worse than images recovered using specialized datasets. See Appendix F for the recovered images. This observation reveals the potential *trade-off* of our research. Overall, training generative models using a general dataset without the knowledge of images classes seems "unconventional." A generative model is typically trained using dataset of only one class (Guo et al., 2019), or it requires image class information to be explicitly provided during both training and generation phases (Brock et al., 2019). Nevertheless, we still evaluate our approach using a general dataset to explore the *full potential* of our attack. We attribute the adequate results using general datasets to discriminable features extracted by our trace encoder from images of different classes. See Appendix G for further evaluations.

From a holistic perspective, the adopted training image sets constitute a *predictive model of user privacy*. While a particular user input is private, we assume that the functionality of victim software (e.g., a human face analysis service) is usually known to the public or can be probed prior to attacks.

## 6 DISCUSSION

This is the first paper that provides a practical solution to reconstruct images from system side channels. Proposing a novel generative model design is *not* our key focus. Also, despite the encouraging results, the reconstructed images show room for improvement. For instance, not all image colors were well recovered. Our manual inspection shows that compared with libjpeg, uPNG does not impose informative side channel dependency on pixel colors (i.e., different colors can likely induce identical side channel logs). Nevertheless, user privacy is presumably leaked as long as the image skeleton is recovered. Colors (or other details) can be recovered if the system community discovered more powerful (finer-grained) side channels.

## 7 Related Work

**Exploiting System Side Channels.** System side channels have been used to exploit various real-life software systems (Dong et al., 2018; Wu et al., 2012; Hähnel et al., 2017; Xu et al., 2015). The CPU cache is shown to be a rich source for SCA attacks on cloud computing environments and web browsers (Hähnel et al., 2017; Oren et al., 2015). In addition to cache line side channels analyzed in this research, other cache storage units, including cache bank and cache set, are also leveraged for SCA attacks (Yarom et al., 2017; Liu et al., 2015a). Overall, while most attacks in this field perform dedicated SCA attacks toward specific side channels, our approach is general and orthogonal to particular side channels.

**Profiling-based SCA.** Machine learning techniques have substantially boosted profiling-based SCA by learning from historical data. DNN models have been used to recover secret keys from crypto libraries under different scenarios (Heuser & Zohner, 2012; Maghrebi et al., 2016; Cagli et al., 2017; Hettwer et al., 2018; Kim et al., 2019; Hettwer et al., 2020). Nevertheless, the success of existing AI-based SCA attacks is primarily driven by the model classification capability, e.g., deciding whether a particular bit of AES secret key is 0 or 1. This paper advocates the new focus on reconstructing images with generative models, leveraging another major breakthrough in DNN.

**SCA Mitigation.** Existing SCA mitigation techniques can be categorized into system-based and software-based approaches. For system-based approaches, previous works have proposed to randomize the cache storage units or enforce fine-grained isolation schemes (Wang & Lee, 2006; 2008; 2007; Liu et al., 2016). Some recent advances propose to leverage new hardware features to mitigate side channel attacks (Gruss et al., 2017). In addition, software-level approaches, including designing secret-independent side channel accesses, randomizing memory access patterns (Coppens et al., 2009; Raj et al., 2009; Schwarz et al., 2018), have also been proposed. Compared with system- and hardware-based mitigations, software-based approaches usually do not require a customized hardware design, and are generally more flexible. Nevertheless, software-based approaches can usually incur extra performance penalty.

## 8 Conclusion

This paper has presented a general and effective SCA framework. The framework is trained with side channels to exploit media software like `libjpeg` and uPNG. Our evaluation shows that reconstructed images manifest close similarity with user inputs, making privacy leakage attacks practical. We also show surprising findings that enabled by our framework, attacks with low-resolution side channels become feasible.

## 9 Ethics Statement

We present a systematic and effective pipeline of recovering private images using system side channels. It is generally acknowledged that studying attack schemes helps eliminate false trust on modern computing infrastructures and promote building secure systems (Athalye et al., 2018). While there is a risk that SCA could become easier using our methods, we believe that our work will also promote rapidly detecting SCA before security breaches. As will be shown in Appendix J, our proposed technique can serve as a "bug detector" to isolate certain code blocks in image processing software that induce SCA opportunities. Developers can thus refer to our findings to patch their software.

Our efforts could impact the ever-growing CV community in building side channel-free image analysis tools. Despite the algorithm-level efforts to address privacy concerns, e.g., via differential privacy (Dwork et al., 2006), the *infrastructure-level* vulnerabilities have not yet received enough attention, especially in the real-world scenarios like MLaaS. Our research will serve as a critical incentive to re-think tradeoffs (e.g., cost vs. security guarantee) currently taken in this field.

## 10 Acknowledgements

We thank the ICLR anonymous reviewers and area chairs for their valuable feedback. Junping Zhang is supported by National Key Research and Development Project (2018YFB1305104).

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

## A   ATTACK SETUP DETAILS

In this section, we provide detailed information regarding our attack setup, including three employed side channels, the attacked `libjpeg` and `uPNG` libraries (libjpeg, 2020; Middleditch, 2010), and how we log side channel information. Three side channels are taken into account as follows:

- **Cache Line.** Cache line side channel denotes one popular hardware side channel, enabling the exploitation of real-world crypto, image and text libraries (Hähnel et al., 2017; Yarom & Falkner, 2014). The CPU cache, as one key component in modern computer architecture, stores data so that future memory requests of that particular data become much faster. Data are stored in a cache block of fixed size, called the cache line. Each memory access made by victim software is projected to a cache line access. In typical cloud platforms, an attacker can monitor cache line accesses made by victim software, leading to a powerful side channel. For modern Intel architectures, the cache line index of a memory address $addr$ can be computed as $addr \gg 6$. Therefore, access of a particular cache line can be mapped back to $2^6$ memory addresses.

- **Page Table.** The OS kernel uses the page table to track mappings between virtual and physical memory addresses. Every memory access made by the victim software is converted into its physical address by querying a page table entry. In cloud computing platforms, a malicious OS on the host can observe page table accesses made by the victim software to infer its memory access (Xu et al., 2015). Given a virtual address $addr$, we calculate the accessed page table index by masking $addr$ with PAGE_MASK $M$: $addr \ \& \ (\sim M)$. $M$ is 4095 on modern x86 architectures (pag, 2020).

- **Data Read/Write Access.** Our preliminary study shows surprising results that enabled by powerful generative models, low-resolution side channels of only one-bit data read/write access records can be used to recover quality images. That is, given a memory access made by the victim software, we use one bit to note whether the access is a read or write operation. Such data read/write accesses can be easily observed by monitoring either cache or page table accesses.

SCA attacks using cache lines and page tables are well-known and have enabled real-life exploitations in various scenarios. In contrast, to our knowledge, no real-world attacks are designed to exploit read/write patterns. This work shows that quality images can be synthesized by using such low-resolution read/write access side channels.

**Preparing Victim Software**   Consistent with existing SCA of exploiting media software (Xu et al., 2015; Hähnel et al., 2017), we use a widely-used JPEG image processing library, `libjpeg`, as the attack target. Attacking `libjpeg` which has been exploited in the literatures makes it easier to (conceptually) compare our approach with existing works. As mentioned in our paper, we indeed contact authors of both papers to inquire their tools; we didn't receive any response by the time of writing. As disclosed in their papers, *manual efforts* are primarily used to recover images. On the other hand, there is **no** issue for our approach to analyze other image processing libraries as long as different inputs adequately influence side channel logs. To demonstrate the *generalizability* of our approach, we also attacked another image library, `uPNG`, which takes images of PNG format as the inputs.

JPEG and PNG are two popular image compression standards. Given an image of JPEG/PNG formats, both image processing libraries reverse the compression step to generate a bitmap image as the prerequisite of many image analysis applications. The decompression process introduces considerable amount of input-dependent side channel accesses for both libraries. We compile both `libjpeg` and `uPNG` on 64-bit Ubuntu 18.04 machine with `gcc` with optimization-level as `-O0` which disables all optimizations.

We measure the complex of `libjpeg`, by counting the line of code of the attacked `libjpeg` module. We report the attacked module, conducting JPEG image decompression under various settings, has approximately 46K lines of code. Similarly, the `uPNG` software has about 1.2K lines of code. In contrast, typically the crypto software attacked by previous profiling-based SCA (Hettwer et al., 2020; Gullasch et al., 2011; Aciicmez & Koc, 2006; Yarom et al., 2017) are much simpler. For

instance, the x86 implementation of Advanced Encryption Standard (AES) in `OpenSSL` has about 600 lines of code (excluding global data structures like permutation boxes).

**Preparing Side Channel Logs** To prepare side channel traces, we use `Pin` (Luk et al., 2005), a runtime monitoring framework developed by Intel to intercept all memory accesses of our test programs when processing an input image $i$. Every virtual address $addr$ on the logged trace is translated into its corresponding cache line and page table indexes following the aforementioned methods. Similarly, we intercept all memory accesses, and for each memory access, we use one bit to denote whether it is a data read or write access. All these runtime monitoring tasks can be done by writing two plugins of `Pin`. We report that processing each image with `libjpeg` can generate a trace of 730K to 760K side channel records. Processing an image with `uPNG` can generate a trace of about 1.3M side channel records. Recall as introduced in Sec. 3.1, each side channel trace is encoded into a $N \times N \times K$ matrix and then processed by CNNs. A `libjpeg` trace is encoded into a $512 \times 512 \times 3$ matrix. A `uPNG` trace is encoded into a $512 \times 512 \times 6$ matrix. We do zero padding for matrices. In comparison, exploiting crypto libraries (e.g., AES decryption) generates much succinct side channel traces with only a few hundred records (Hettwer et al., 2020),

**Attacking Other Image Processing Software and DNN Models** We pick `libjpeg` since this is the *only* media software attacked by existing SCA (Xu et al., 2015; Hähnel et al., 2017). We also attacked `uPNG` to demonstrate the generalizability of our approach. Note that `libjpeg` is commonly used in the image analysis pipeline, e.g., it is the prerequisite of the popular Python image processing library — `Pillow` (Clark, 2020).

Also, careful readers may wonder the feasibility of directly exploiting DNN-based image analysis tools. However, as pointed out in previous research (Hong et al., 2018a), memory access of typical DNN operations like matrix multiplications is **not** input-dependent. That is, while it has been demonstrated by the same authors that cache side channels is feasible to recover DNN model architectures (Hong et al., 2020), SCA is generally **not** feasible to recover inputs to DNN models.

## B  MODEL ARCHITECTURE AND EXPERIMENT SETUP

We now report the architecture and parameters of our framework. The $Enc_{trace}$ of VAE-LP module is reported in Table 5. $Enc_{image}$ and $Dec_{image}$ are reported in Table 4. The generator $G$ and discriminator $D$ of our GAN module are listed in Table 6 and Table 7, respectively.

We implement our framework in Pytorch (ver. 1.5.0). We use the Adam optimizer (Kingma & Ba, 2014) with learning rate $\eta_{VAE-LP} = 0.0001$ for the VAE-LP module, and learning rate $\eta_{GAN} = 0.0002$ for the GAN module. We set $\beta_1 = 0.5$, and $\beta_2 = 0.999$ for both modules. $\beta$ in $Loss_{VAE-LP}$ is 0.0001, and $\gamma$ in $Loss_{GAN}$ is 100. Minibatch size is 50.

We ran our experiments on an Intel Xeon CPU E5-2678 with 256 GB of RAM and one Nvidia GeForce RTX 2080 GPU. The training is completed at 200 iterations (100 iterations for the VAE-LP module, and 100 iterations for the GAN module).

Table 4: The image encoder $Enc_{image}$ and image decoder $Dec_{image}$ architectures.

| Image Encoder | | | Image Decoder | | |
|---|---|---|---|---|---|
| Layer | Act. | Output shape | Layer | Act. | Output shape |
| Input | - | $3 \times 128 \times 128$ | Input | - | $128 \times 1 \times 1$ |
| Conv $4 \times 4_{/2,1}$ | LReLU(0.2) | $128 \times 64 \times 64$ | Deconv $4 \times 4_{/1,0}$ | BN + ReLU | $128 \times 4 \times 4$ |
| ResBlock (He et al., 2016) | - | $128 \times 64 \times 64$ | ResBlock | - | $128 \times 4 \times 4$ |
| Conv $4 \times 4_{/2,1}$ | BN + LReLU(0.2) | $128 \times 32 \times 32$ | Deconv $4 \times 4_{/2,1}$ | BN + ReLU | $128 \times 8 \times 8$ |
| ResBlock | - | $128 \times 32 \times 32$ | ResBlock | - | $128 \times 8 \times 8$ |
| Conv $4 \times 4_{/2,1}$ | BN + LReLU(0.2) | $128 \times 16 \times 16$ | Deconv $4 \times 4_{/2,1}$ | BN + ReLU | $128 \times 16 \times 16$ |
| ResBlock | - | $128 \times 16 \times 16$ | ResBlock | - | $128 \times 16 \times 16$ |
| Conv $4 \times 4_{/2,1}$ | BN + LReLU(0.2) | $128 \times 8 \times 8$ | Deconv $4 \times 4_{/2,1}$ | BN + ReLU | $128 \times 32 \times 32$ |
| ResBlock | - | $128 \times 8 \times 8$ | ResBlock | - | $128 \times 32 \times 32$ |
| Conv $4 \times 4_{/2,1}$ | BN + LReLU(0.2) | $128 \times 4 \times 4$ | Deconv $4 \times 4_{/2,1}$ | BN + ReLU | $128 \times 64 \times 64$ |
| ResBlock | - | $128 \times 4 \times 4$ | ResBlock | - | $128 \times 64 \times 64$ |
| Conv $4 \times 4_{/1,0}$ | LReLU(0.2) | $128 \times 1 \times 1$ | Deconv $4 \times 4_{/2,1}$ | Tanh | $3 \times 128 \times 128$ |
| Output | - | $128$ | Output | - | $3 \times 128 \times 128$ |

Table 5: The trace encoder $Enc_{trace}$ architecture.

| Trace Encoder | | |
|---|---|---|
| Layer | Act. | Output shape |
| Input | - | $3 \times 512 \times 512$ |
| Normalize | - | $3 \times 512 \times 512$ |
| Conv $4 \times 4_{/2,1}$ | BN + LReLU(0.2) | $64 \times 256 \times 256$ |
| Conv $4 \times 4_{/2,1}$ | BN + LReLU(0.2) | $64 \times 128 \times 128$ |
| Conv $4 \times 4_{/2,1}$ | BN + LReLU(0.2) | $64 \times 64 \times 64$ |
| Conv $4 \times 4_{/2,1}$ | BN + LReLU(0.2) | $128 \times 32 \times 32$ |
| Conv $4 \times 4_{/2,1}$ | BN + LReLU(0.2) | $256 \times 16 \times 16$ |
| Conv $4 \times 4_{/2,1}$ | BN + LReLU(0.2) | $512 \times 8 \times 8$ |
| Conv $4 \times 4_{/2,1}$ | BN + LReLU(0.2) | $512 \times 4 \times 4$ |
| Conv $4 \times 4_{/1,0}$ | BN + Tanh | $128 \times 1 \times 1$ |
| Output | - | $128$ |

Table 6: The generator $G$ architecture.

| Encoder | | | Decoder | | |
|---|---|---|---|---|---|
| Layer | Act. | Output shape | Layer | Act. | Output shape |
| Input | - | $3 \times 128 \times 128$ | Input | - | $1024 \times 1 \times 1$ |
| Conv $4 \times 4_{/2,1}$ | - | $128 \times 64 \times 64$ | Deconv $4 \times 4_{/2,1}$ | BN + ReLU | $1024 \times 2 \times 2$ |
| Conv $4 \times 4_{/2,1}$ | BN + LReLU(0.2) | $256 \times 32 \times 32$ | Deconv $4 \times 4_{/2,1}$ | BN + ReLU | $1024 \times 4 \times 4$ |
| Conv $4 \times 4_{/2,1}$ | BN + LReLU(0.2) | $512 \times 16 \times 16$ | Deconv $4 \times 4_{/2,1}$ | BN + ReLU | $1024 \times 8 \times 8$ |
| Conv $4 \times 4_{/2,1}$ | BN + LReLU(0.2) | $1024 \times 8 \times 8$ | Deconv $4 \times 4_{/2,1}$ | BN + ReLU | $512 \times 16 \times 16$ |
| Conv $4 \times 4_{/2,1}$ | BN + LReLU(0.2) | $1024 \times 4 \times 4$ | Deconv $4 \times 4_{/2,1}$ | BN + ReLU | $256 \times 32 \times 32$ |
| Conv $4 \times 4_{/2,1}$ | BN + LReLU(0.2) | $1024 \times 2 \times 2$ | Deconv $4 \times 4_{/2,1}$ | BN + ReLU | $128 \times 64 \times 64$ |
| Conv $4 \times 4_{/2,1}$ | LReLU(0.2) | $1024 \times 1 \times 1$ | Deconv $4 \times 4_{/2,1}$ | Tanh | $3 \times 128 \times 128$ |
| Output | - | $1024 \times 1 \times 1$ | Output | - | $3 \times 128 \times 128$ |

Table 7: The discriminator $D$ architecture.

| Discriminator | | |
|---|---|---|
| Layer | Act. | Output shape |
| Input | - | $3 \times 128 \times 128$ |
| Conv $4 \times 4_{/2,1}$ | BN + LReLU(0.2) | $128 \times 64 \times 64$ |
| Conv $4 \times 4_{/2,1}$ | BN + LReLU(0.2) | $256 \times 32 \times 32$ |
| Conv $4 \times 4_{/2,1}$ | BN + LReLU(0.2) | $512 \times 16 \times 16$ |
| Conv $4 \times 4_{/2,1}$ | BN + LReLU(0.2) | $1024 \times 8 \times 8$ |
| Conv $4 \times 4_{/2,1}$ | BN + LReLU(0.2) | $1024 \times 4 \times 4$ |
| Conv $4 \times 4_{/1,0}$ | Sigmoid | $1 \times 1 \times 1$ |
| Output | - | $1$ |

## C SAMPLE OUTPUTS WHEN ATTACKING LIBJPEG

This section provides more images generated by our framework when attacking libjpeg. Overall, we interpret the results as promising and highly consistent across all three different datasets. As discussed in Sec. 5.1, the reconstructed and the corresponding reference images show highly consistent identities, such as gender, face orientation, human gesture, and hair style.

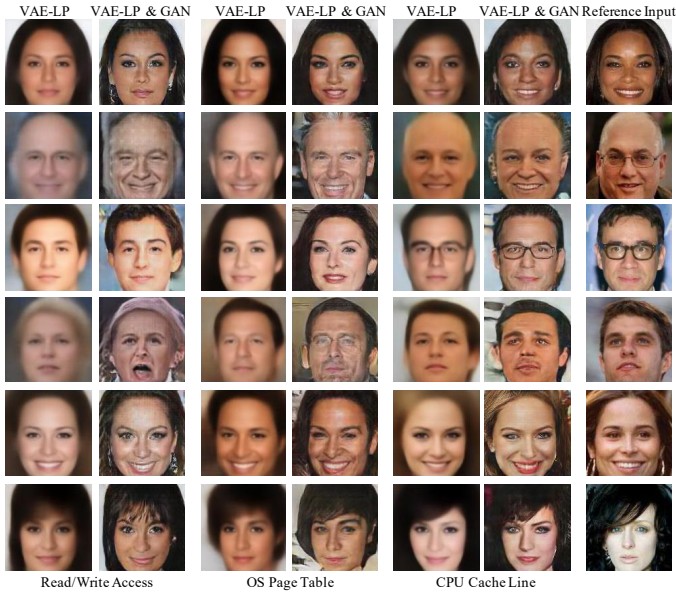

Figure 5: Evaluation results of CelebA dataset in terms of three different side channels. The reference images (i.e., private user inputs) are presented on the right most column for the reference.

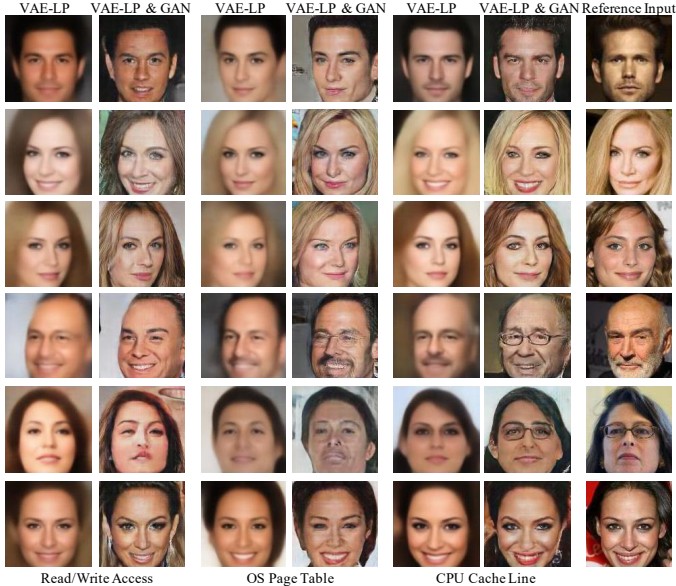

Figure 6: Evaluation results of CelebA dataset in terms of three different side channels. The reference images (i.e., private user inputs) are presented on the right most column for the reference.

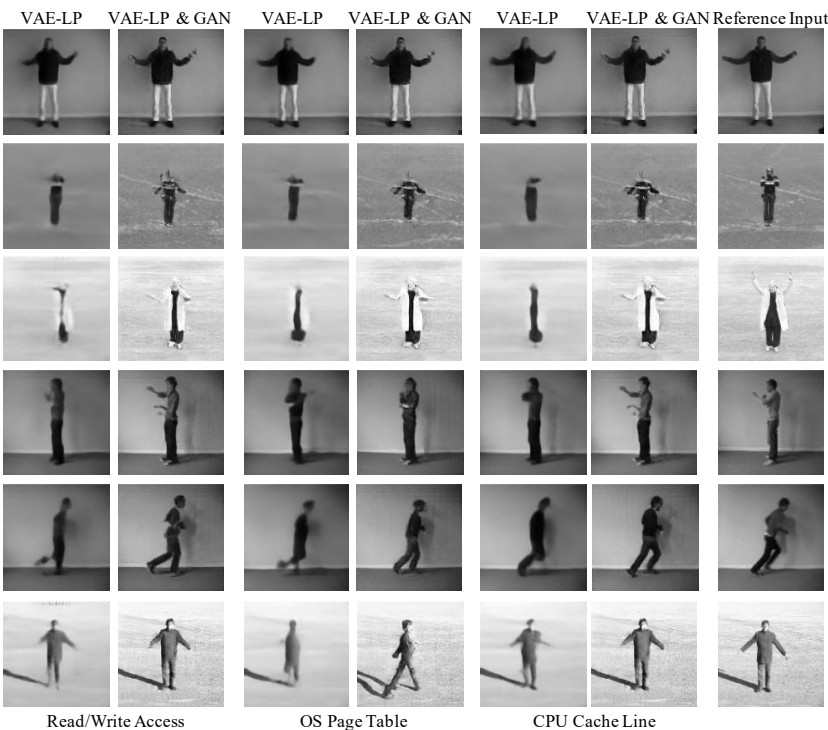

Figure 7: Evaluation results of KTH dataset in terms of three different side channels. The reference images (i.e., private user inputs) are presented on the right most column for the reference.

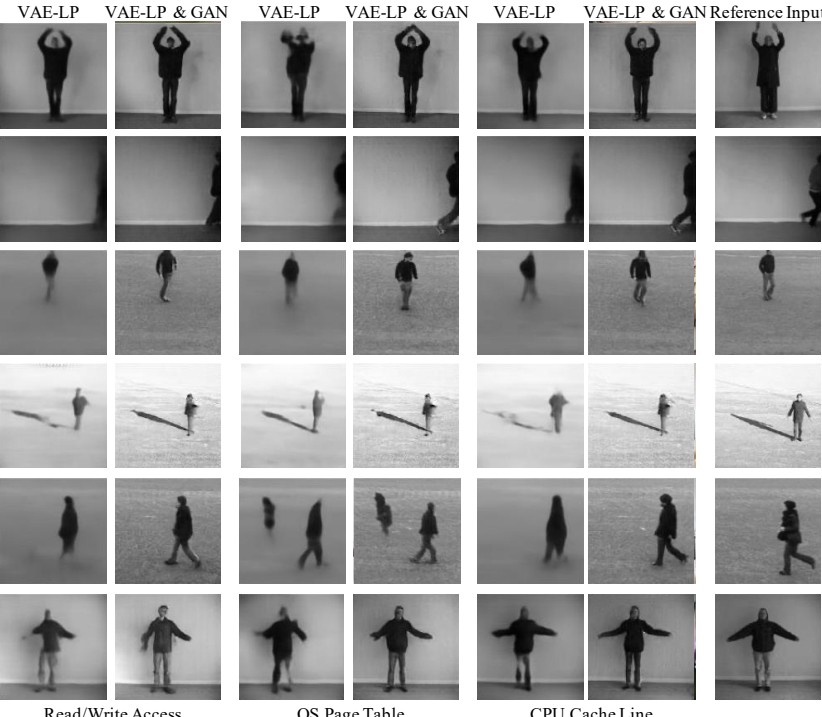

Figure 8: Evaluation results of KTH dataset in terms of three different side channels. The reference images (i.e., private user inputs) are presented on the right most column for the reference.

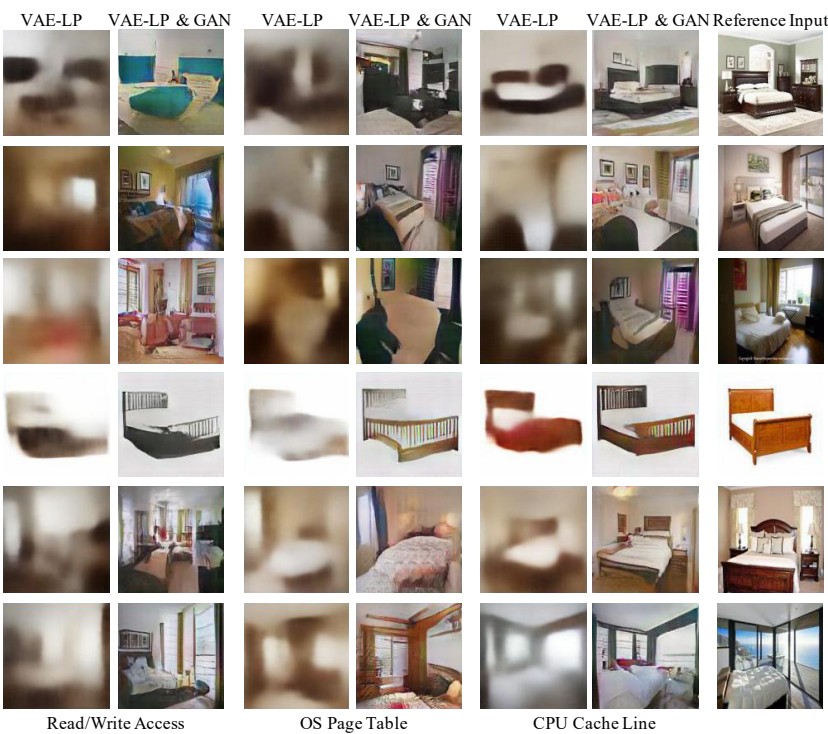

Figure 9: Evaluation results of LSUN dataset in terms of three different side channels. The reference images (i.e., private user inputs) are presented on the right most column for the reference.

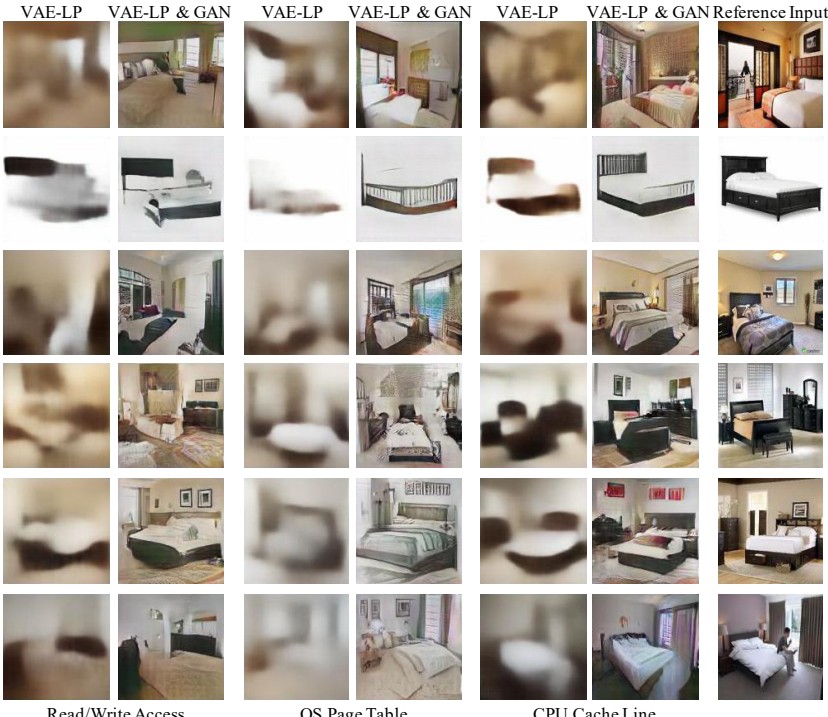

Figure 10: Evaluation results of LSUN dataset in terms of three different side channels. The reference images (i.e., private user inputs) are presented on the right most column for the reference.

# D  QUANTITATIVE EVALUATION

Besides the quantitative evaluation of the discriminability of reconstructed images reported in the paper, we also analyze images reconstructed by using only the VAE-LP module and presented the results in Table 8. Accordingly, we give the quantitative data that has been already reported in our paper for cross comparison in Table 9.

Table 8: Discriminability evaluation results on outputs using VAE-LP (higher is better). We also present the baseline, random guess (RG), for comparison.

|  | $k = 1\ N = 100$ | | | | $k = 1\ N = 500$ | | | |
|---|---|---|---|---|---|---|---|---|
|  | CelebA | KTH | LSUN | RG | CelebA | KTH | LSUN | RG |
| Cache Line | 49.98% | 30.28% | 10.36% | 1% | 31.18% | 19.82% | 3.30% | 0.2% |
| Page Table | 14.66% | 29.00% | 1.94% | 1% | 5.98% | 18.20% | 0.50% | 0.2% |
| Read/Write | 17.52% | 24.46% | 1.98% | 1% | 7.86% | 14.92% | 0.40% | 0.2% |
|  | $k = 5\ N = 100$ | | | | $k = 5\ N = 500$ | | | |
|  | CelebA | KTH | LSUN | RG | CelebA | KTH | LSUN | RG |
| Cache Line | 78.00% | 54.42% | 29.74% | 5% | 56.06% | 32.86% | 11.50% | 1% |
| Page Table | 35.38% | 52.70% | 7.92% | 5% | 16.92% | 31.54% | 1.92% | 1% |
| Read/Write | 39.16% | 47.06% | 7.08% | 5% | 19.68% | 26.02% | 1.86% | 1% |
|  | $k = 20\ N = 100$ | | | | $k = 20\ N = 500$ | | | |
|  | CelebA | KTH | LSUN | RG | CelebA | KTH | LSUN | RG |
| Cache Line | 91.98% | 79.16% | 59.66% | 20% | 75.98% | 51.20% | 26.72% | 4% |
| Page Table | 61.94% | 76.88% | 24.90% | 20% | 32.96% | 50.32% | 6.48% | 4% |
| Read/Write | 65.46% | 72.92% | 24.56% | 20% | 36.90% | 43.78% | 6.30% | 4% |

Table 9: Discriminability evaluation results on the outputs using VAE-LP & GAN modules (higher is better). We have already reported these results in Table 2.

|  | $k = 1\ N = 100$ | | | | $k = 1\ N = 500$ | | | |
|---|---|---|---|---|---|---|---|---|
|  | CelebA | KTH | LSUN | RG | CelebA | KTH | LSUN | RG |
| Cache Line | 47.28% | 38.22% | 10.00% | 1% | 28.68% | 24.50% | 3.04% | 0.2% |
| Page Table | 16.16% | 35.86% | 2.26% | 1% | 6.56% | 22.58% | 0.46% | 0.2% |
| Read/Write | 19.58% | 33.44% | 2.24% | 1% | 8.42% | 20.64% | 0.50% | 0.2% |
|  | $k = 5\ N = 100$ | | | | $k = 5\ N = 500$ | | | |
|  | CelebA | KTH | LSUN | RG | CelebA | KTH | LSUN | RG |
| Cache Line | 74.98% | 63.56% | 28.74% | 5% | 52.22% | 41.24% | 10.90% | 1% |
| Page Table | 37.96% | 61.28% | 8.62% | 5% | 17.90% | 38.80% | 2.42% | 1% |
| Read/Write | 42.44% | 58.30% | 8.28% | 5% | 21.18% | 36.22% | 2.54% | 1% |
|  | $k = 20\ N = 100$ | | | | $k = 20\ N = 500$ | | | |
|  | CelebA | KTH | LSUN | RG | CelebA | KTH | LSUN | RG |
| Cache Line | 90.70% | 83.72% | 58.88% | 20% | 72.98% | 61.08% | 25.92% | 4% |
| Page Table | 65.14% | 82.22% | 26.42% | 20% | 35.22% | 58.90% | 7.30% | 4% |
| Read/Write | 67.82% | 80.14% | 26.64% | 20% | 39.84% | 55.74% | 7.08% | 4% |

Comparing results reported in Table 8 and Table 9, we observed that by using VAE-LP & GAN modules together, the KTH human gesture dataset has a substantial improvement in terms of accuracy. The average accuracy of the KTH dataset is 41.9% in Table 8 while the average accuracy of the KTH dataset is increased to 49.8% in Table 9. This observation is consistent with our findings in Sec. 5.2 and some cases demonstrated in Fig. 7 and Fig. 8. Recall we observed an obscure "black bar" in KTH images reconstructed by only using VAE-LP module, while a human gesture can be clearly identified by enhancing the "black bar" with the GAN module. We also observed improved accuracy for the CelebA (41.0% to 41.4%) and LSUN datasets (12.6% to 12.9%) when supplementing VAE-LP with GAN. Overall, we interpret that better discriminability can be achieved when using GAN, implying higher success rate for attackers to de-anonymize user identity and privacy.

We also present fine-grained facial attribute comparison between the reconstructed images and reference inputs. In Sec. 5.2 we have reported gender and age confusion matrices evaluation using cache side channels (also presented in Fig. 11 for the reference and cross comparison), we also report other settings in Fig. 12, Fig. 13, and Fig. 14. To quantitatively evaluate the confusion matrices, we measure and report the weighted-average F1 score in Table 10. Besides one case with notably increased F1 score (the gender matrix using page table), VAE-LP and VAE-LP & GAN have comparable weighted-average F1 scores.

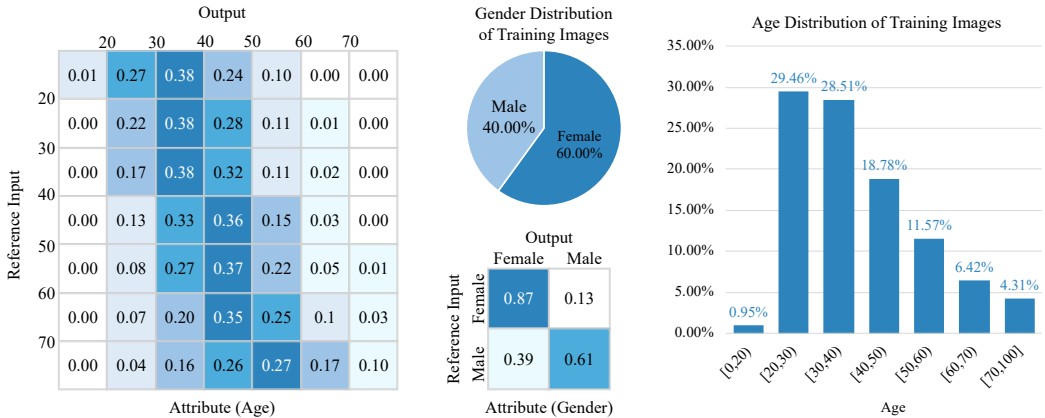

Figure 11: Facial attribute evaluation in terms of cache side channels. Images are generated using VAE-LP & GAN modules. Confusion matrices (with row-wise normalization) compare the age and gender classification results on our recovered images and those obtained from reference inputs. We also report age and gender distribution in the dataset for comparison. We have already reported these results in Fig. 4.

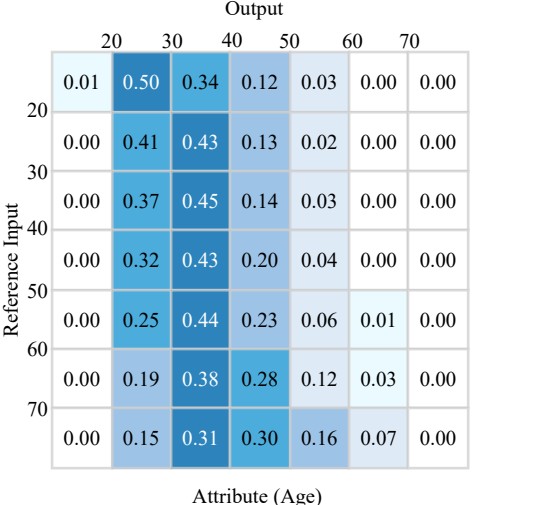

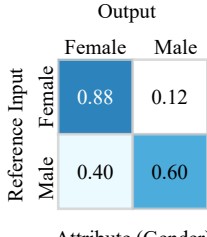

Figure 12: Facial attribute evaluation in terms of cache side channels. Images are generated using the VAE-LP module. Confusion matrices (with row-wise normalization) compare the age and gender classification results on our recovered images and those obtained from reference inputs. The age and gender distributions in the dataset have been given in Fig. 11.

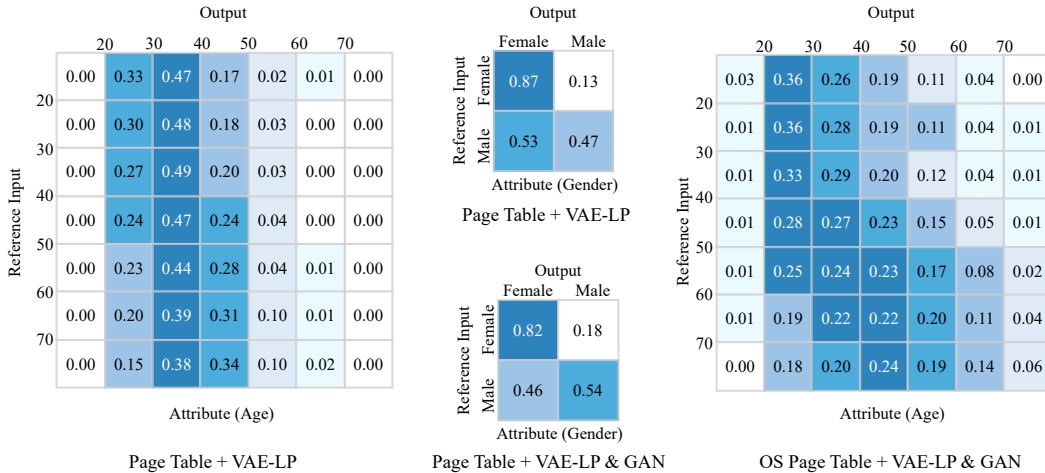

Figure 13: Facial attribute evaluation in terms of page table side channels. We present evaluation results by using images reconstructed using only VAE-LP module, and images reconstructed using VAE-LP & GAN. Confusion matrices (with row-wise normalization) compare the age and gender classification results on our recovered images and those obtained from reference inputs. The age and gender distribution in the dataset have been given in Fig. 11.

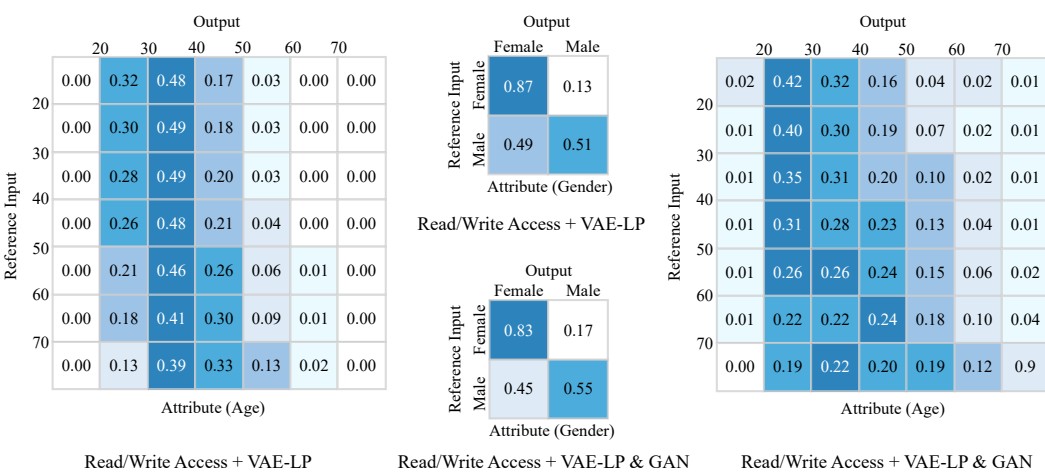

Figure 14: Facial attribute evaluation in terms of memory read/write access pattern side channels. We present evaluation results by using images reconstructed using only VAE-LP module, and images reconstructed using VAE-LP & GAN. Confusion matrices (with row-wise normalization) compare the age and gender classification results on our recovered images and those obtained from reference inputs. The age and gender distribution in the dataset have been given in Fig. 11.

Table 10: Weighted-average F1-score (higher is better).

|  | Age | | Gender | |
|---|---|---|---|---|
|  | VAE-LP | VAE-LP & GAN | VAE-LP | VAE-LP & GAN |
| CPU Cache Line | 0.27 | 0.27 | 0.76 | 0.75 |
| OS Page Table | 0.25 | 0.26 | 0.68 | 0.79 |
| Memory Read/Write | 0.25 | 0.27 | 0.71 | 0.71 |

# E  SAMPLE OUTPUTS WHEN ATTACKING uPNG

This section provides images generated by our framework when attacking uPNG. While the side channel traces induced by uPNG is generally less informative than libjpeg (as shown in Table 3 and discussed in Sec. 6), we still observed high visually consistent identities between the reconstructed images and their reference inputs, including gender, face orientation, hair style, hair length, whether wearing a pair of glasses and many other factors.

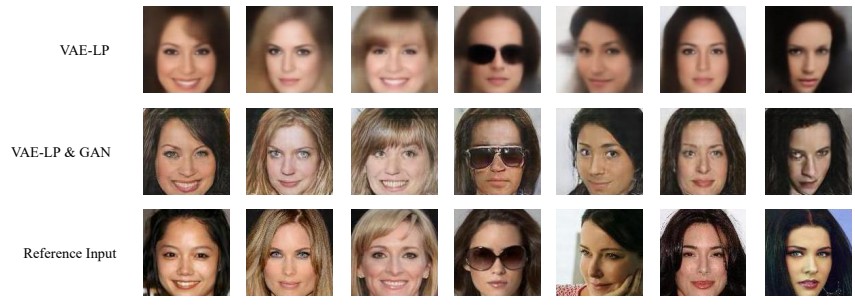

Figure 15: Evaluation results of CelebA dataset in terms of the cache side channels.

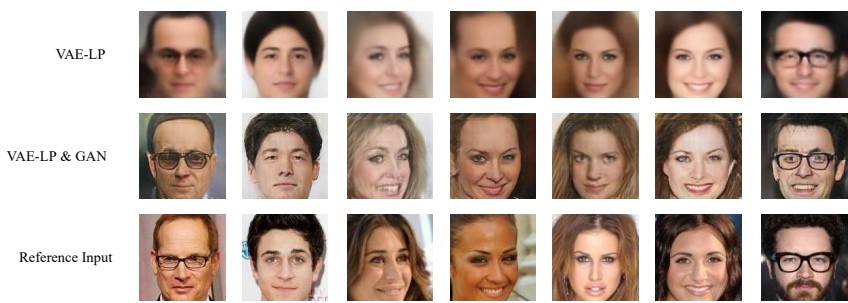

Figure 16: Evaluation results of CelebA dataset in terms of the cache side channels.

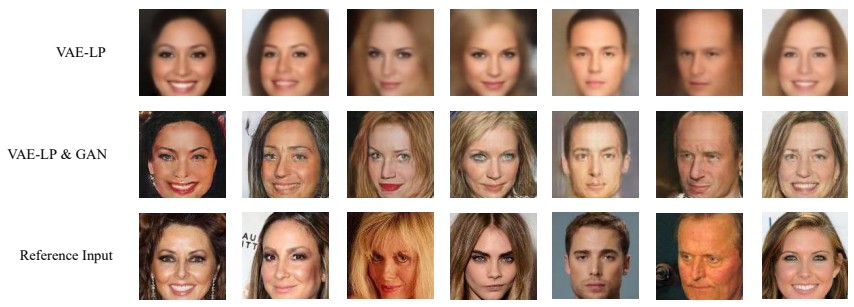

Figure 17: Evaluation results of CelebA dataset in terms of the cache side channels.

## F   SAMPLE OUTPUTS WHEN TRAINING WITH MINI-IMAGENET

To measure our attack reconstructing arbitrary images without knowing the type of images being processed, Sec. 5.3 reports model training and attack performance using a general dataset, mini-Imagenet. We report that the mini-Imagenet dataset has in total 60K images of 100 classes, and we divide each class (with 600 images) into 480 training images and 120 testing images. As a result, we have in total 48K images for training and take the other 12K images for testing. While training generative models with a general dataset is not the common practice unless image class information is explicitly provided (Brock et al., 2019), Table 3 still reports highly encouraging results of the discriminability analysis by largely outperforming the baseline — random guess. In this section, we provide sample images generated at this step.

The synthesized images from the mini-Imagenet dataset is visually much worse that images synthesized from specific datasets (e.g., `CelebA` in Fig. 5). Nevertheless, by comparing the synthesized images and their corresponding references (i.e., user inputs), we interpret that images still exhibit high discriminability, in the sense that many visually consistent image skeletons and colors are recovered, indicating adequate leakage of user privacy.

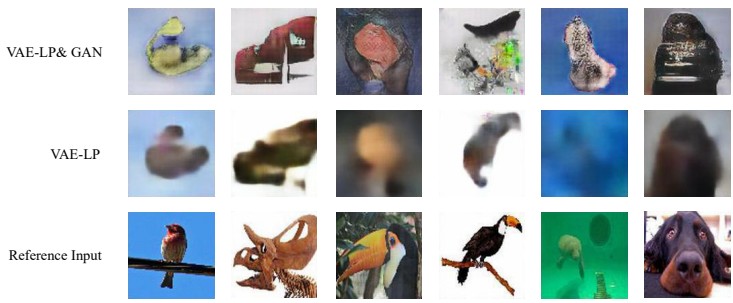

Figure 18: Evaluation results of mini-Imagenet dataset when attacking `libjpeg` with cache side channels.

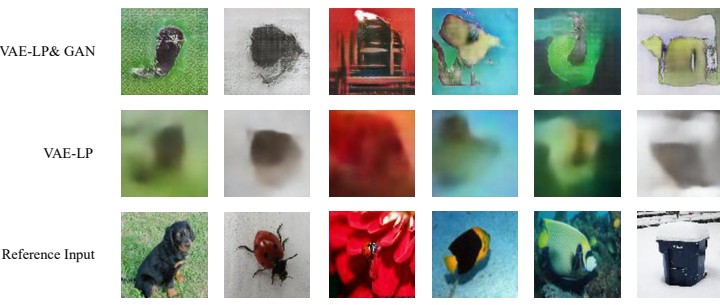

Figure 19: Evaluation results of mini-Imagenet dataset when attacking `libjpeg` with cache side channels.

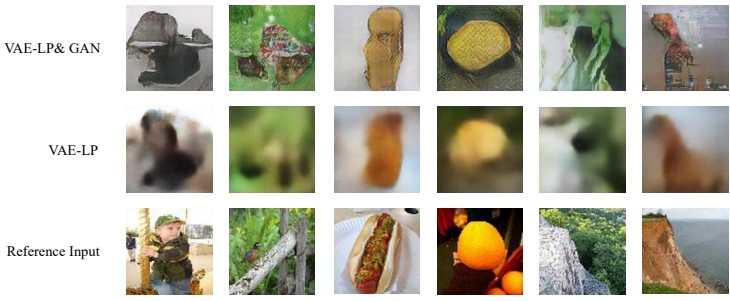

Figure 20: Evaluation results of mini-Imagenet dataset when attacking `libjpeg` with cache side channels.

# G    CLASSIFYING OUTPUTS OF THE TRACE ENCODER

By training our framework with mini-Imagenet and assessing the discriminability, Sec. 5.3 has demonstrated that our attack can largely outperform the baseline even with no prior knowledge on the class information of user private images. We attribute the promising evaluation results to the discriminable features successfully extracted by our trace encoder (trained with mini-Imagenet; see Appendix F). This section presents empirical results by assessing to what extent the latent representations derived from images of two randomly selected classes are distinguishable.

To this end, we build a binary classifier, by piggybacking our trace encoder with a fully-connected (FC) layer and using Sigmoid as the activation function. As mentioned in Appendix F, the mini-Imagenet dataset has in total 60K images of 100 classes, and we divide each class into 480 images for training and 120 images for testing. At this step, we randomly select two classes of images, and use their training sets (in total 960 images) to train the proposed binary classifier. We then use their testing sets, including in total 240 images, to measure the binary classification accuracy. It is worth mentioning that we only train the classifier for one epoch to highlight that the latent representations extracted by the trace encoder already exhibit high quality and discriminability. We only tune the parameters of FC layer but preserve the parameters of our trace encoder.

We iterate this process for 100 times. Fig. 21 reports the classification accuracy across all 100 binary classification tasks. While the baseline accuracy for our binary classification task is 50%, most tasks exhibit much higher classification accuracy. We report the average accuracy is 81.6% and 32 cases exhibit a classification accuracy above 90%.

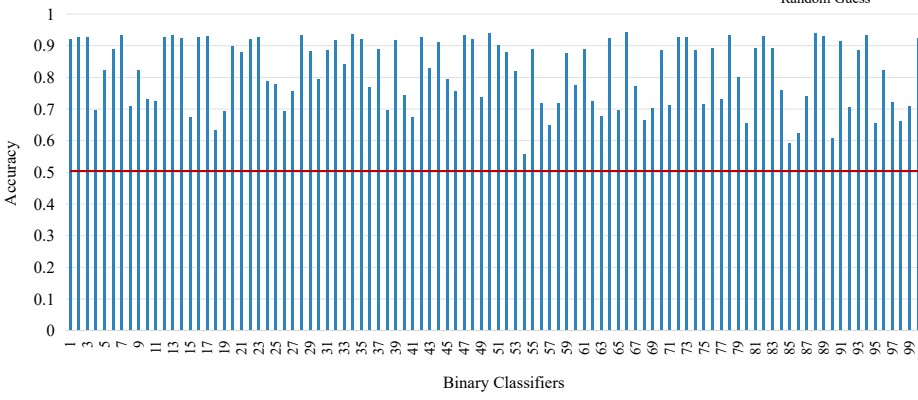

Figure 21: Classification accuracy across 100 binary classifiers.

## H   ROBUSTNESS TO NOISE

This section provides experiences on the robustness of our framework by inserting noise into the side channel traces. To this end, we evaluated the following three settings:

- **Gaussian noise insertion**: For each side channel data point $input$ on the side channel trace, $input = x \times n + (1-x) \times input$, where $x \in [0.1, 0.2, 0.5]$, and $n$ denotes randomly generated noise following the Gaussian Distribution.
- **Zero replacement**: Randomly set $x\%$ of the data points on the side channel trace to 0, where $x \in [10, 20, 50]$.
- **Round shifting**: Round shifting the side channel trace for $x$ steps, where $x \in [1, 10, 100]$.

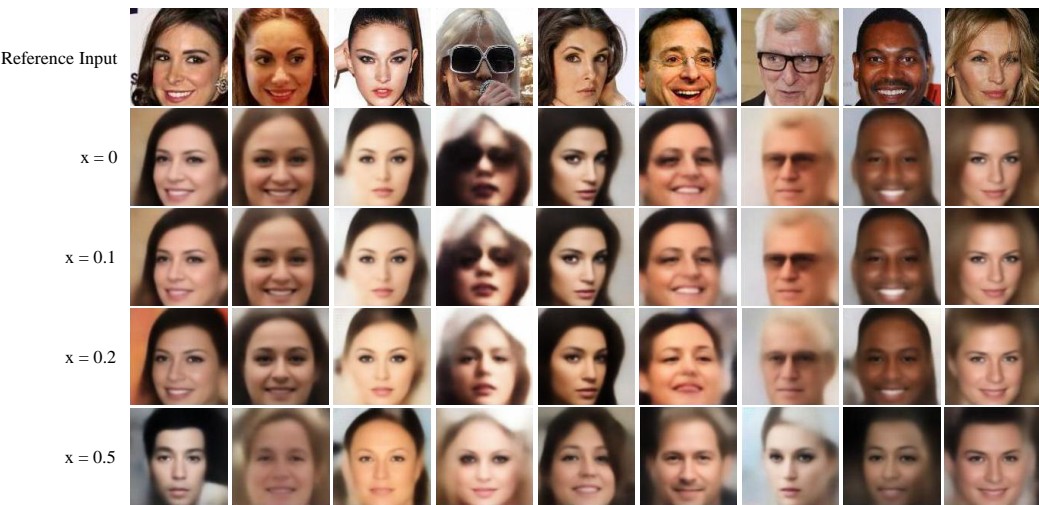

Figure 22: Evaluation results of `CelebA` dataset in terms of cache side channels and three different noise setting. We also present the synthesized images with no noise for the reference. User input images are presented on the first row for the reference.

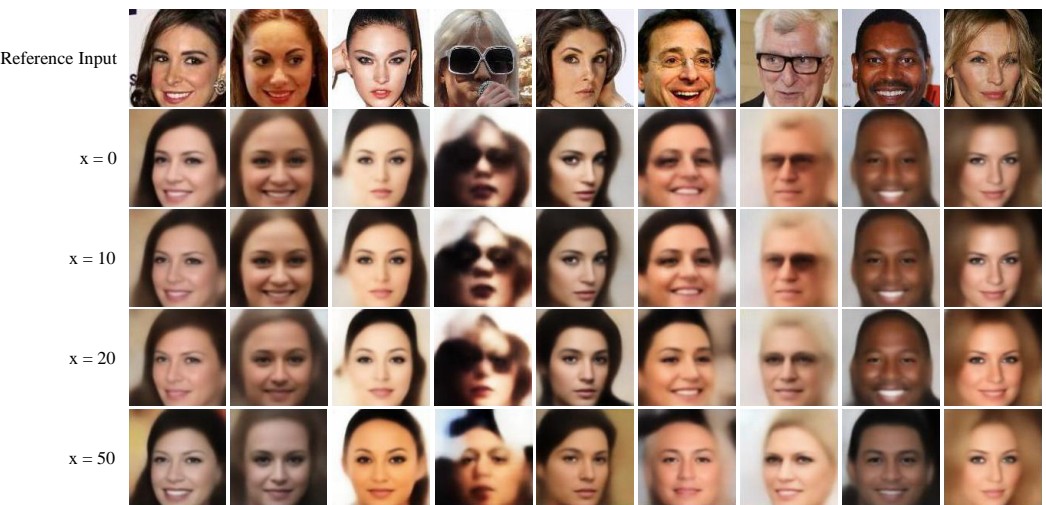

Figure 23: Evaluation results of `CelebA` dataset in terms of cache side channels and three different random zero replacement setting. We also present the synthesized images with no zero replacement ($x = 0$) for the reference. User input images are presented on the first row for the reference.

Table 11: Discriminability evaluation by adding Gaussian noise in the side channel trace.

| Configuration | $x = 0$ | $x = 0.1$ | $x = 0.2$ | $x = 0.5$ |
|---|---|---|---|---|
| $k = 1\ N = 100$ | 49.98% | 48.32% | 39.14% | 5.66% |
| $k = 5\ N = 100$ | 78.00% | 76.28% | 67.08% | 19.62% |
| $k = 20\ N = 100$ | 91.98% | 90.56% | 86.22% | 45.20% |

Table 12: Discriminability evaluation by randomly replacing x% side channel data points with zero.

| Configuration | $x = 0$ | $x = 10$ | $x = 20$ | $x = 50$ |
|---|---|---|---|---|
| $k = 1\ N = 100$ | 49.98% | 45.02% | 38.50% | 14.40% |
| $k = 5\ N = 100$ | 78.00% | 73.30% | 65.52% | 35.50% |
| $k = 20\ N = 100$ | 91.98% | 89.42% | 85.86% | 64.14% |

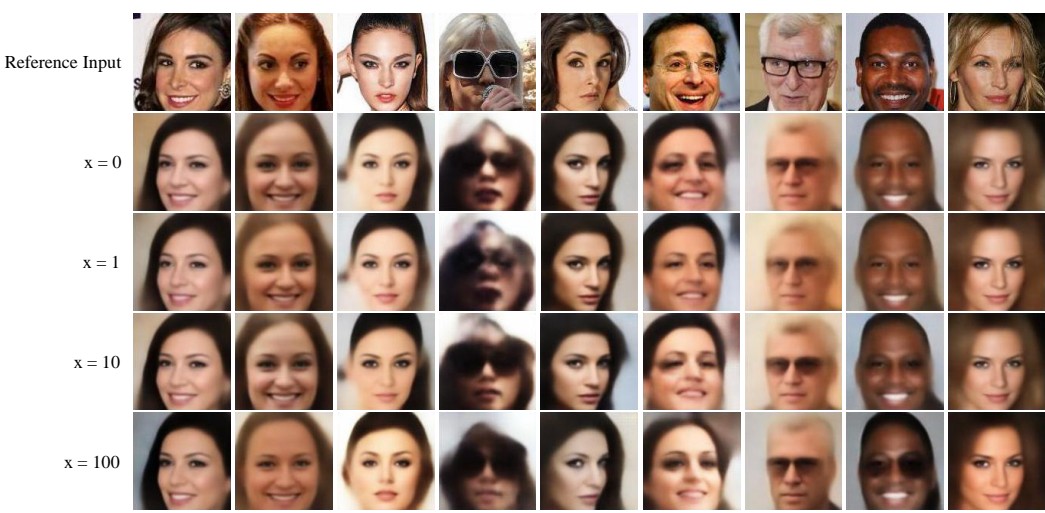

Figure 24: Evaluation results of `CelebA` dataset in terms of cache side channels and three different shifting setting. We also present the synthesized images with no shifting ($x = 0$) for the reference. User input images are presented on the first row for the reference.

We present the corresponding qualitative evaluation results in Fig. 22, Fig. 23, and Fig. 24, respectively. Accordingly, we present the quantitative results in Table 11, Table 12, and Table 13.

Overall, despite the challenging settings, we still observed considerable visually consistent features (e.g., face orientation, hair style, gender) between the reconstructed images and their reference inputs. Fig. 24 shows that round shifting seems to impose relatively low impact on the reconstructed images (e.g., comparing $x = 0$ with $x = 100$). In contrast, a more challenging setting, adding Gaussian noise to each side channel data point, causes observable effect on the constructed images (e.g., comparing $x = 0$ with $x = 0.5$).

Table 13: Discriminability evaluation by round shifting the side channel trace.

| Configuration | $x = 0$ | $x = 1$ | $x = 10$ | $x = 100$ |
|---|---|---|---|---|
| $k = 1\ N = 100$ | 49.98% | 49.56% | 49.94% | 47.48% |
| $k = 5\ N = 100$ | 78.00% | 77.42% | 77.00% | 75.30% |
| $k = 20\ N = 100$ | 91.98% | 91.44% | 91.52% | 90.66% |

## I   ABLATION EXPERIMENTS

This section provides more ablation experiments. We aim to demonstrate the necessity of *image encoder* and *a learned prior*. To this end, we launch experiences to synthesize images without using the image encoder (see the 4th row of Fig. 25), and also synthesize images with a fixed Gaussian prior (the 3rd row of Fig. 25). It is easy to see that the reconstructed images manifest much lower quality compared with images synthesized by our framework (the 2nd row of Fig. 25). In particular, images synthesized without using the image encoder are seen to contain grids (the last row of Fig. 25). It is also observed that when feeding the decoder with a fixed gaussian prior, the synthesized images are low quality as well. The outputs become not recognizable since the fixed prior has primarily no information of side channel traces. This also indicates that our model is not a simple image generator, and the trace encoder plays a vital role in the pipeline. Overall, we interpret these ablation evaluations highlight the importance of the image encoder and a learned prior in SCA. Our designed framework, by incorporating the side channel trace encoder, can effectively extract latent representations from the logged side channel data points. Simultaneously, by integrating corresponding reference images during training, we provide a guideline to help the image decoder to generate quality images.

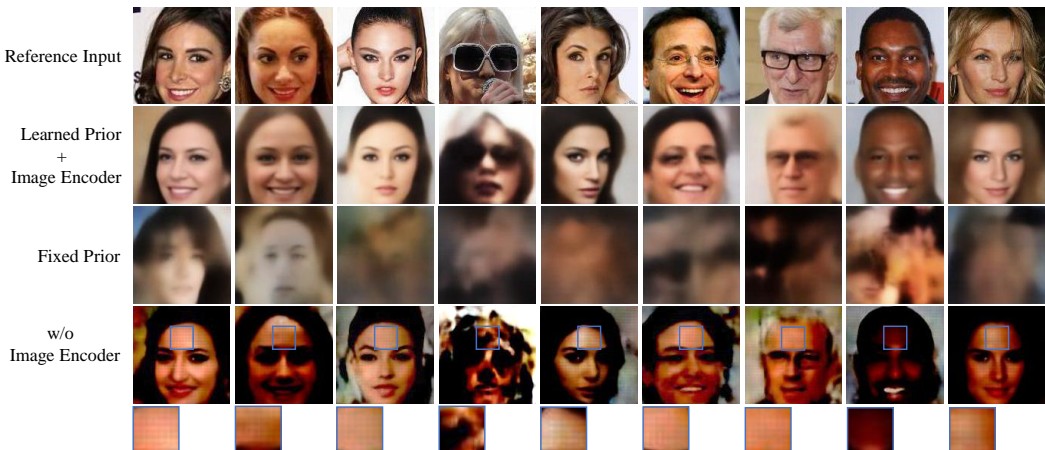

Figure 25: Evaluation results of `CelebA` dataset in terms of cache side channels and different settings. User input images are presented on the first row for the reference. Images synthesized by our presented framework are presented on the second row.

In addition, we further conduct another ablation experiments regarding image level metrics. To do so, we use LPIPS (Zhang et al., 2018), image-level perceptual loss, to calculate the similarity of the reconstructed image and ground truth image.

The results are given in Table 14. Compared with our results reported in Sec. 5, the accuracy of GAN output is reduced by around 10% and reduced even more on output of VAE-LP. Nevertheless, the results are reasonable since the model is not train using this perceptual loss. Overall, we interpret the evaluation results show that from the perspective of "human adversaries", the GAN module is indeed necessary.

Table 14: Ablation experiments with LPIPS.

| Model | $K = 1, N = 100$ | $K = 5, N = 100$ | $K = 20, N = 100$ |
|---|---|---|---|
| VAE-LP | 23.02% | 47.92% | 72.40% |
| VAE-LP & GAN | 39.64% | 63.12% | 81.42% |

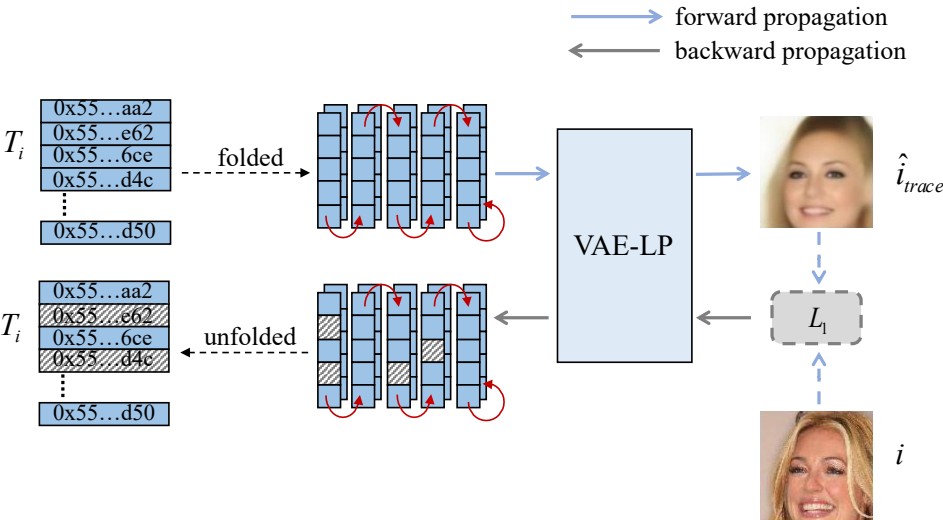

Figure 26: Calculating gradient using backward propagation.

## J  MAPPING SIDE CHANNEL TRACES BACK TO INFORMATIVE FUNCTIONS

In this section, we explore side channel traces and aim to answer the question "what information in the side channel is used for image reconstruction?" To this end, we explore which data points on the side channel trace affects most to the output by calculating the gradient. Due to the limited time, we tested cache side channel traces logged for the `libjpeg` software.

Given an image $i$ which is not in training data, we first collect its corresponding cache side channel trace $T_i$ when being processed by `libjpeg`. VAE-LP module will then take $T_i$ as the input and reconstruct $i_{trace}$. As shown in Fig. 26, we then calculate the loss of $i_{trace}$ and $i$, and further perform backward propagation to calculate the gradient up to $T_i$, namely $g_{T_i}$. Since $g_{T_i}$ has the same size of $T_i$, we can pinpoint which part of $T_i$ contributes most to $i_{trace}$ by localizing which part of $g_{T_i}$ has a higher value. More specifically, we normalize $g_{T_i}$ to $[0, 1]$ and only keep values greater than a threshold $\mathcal{T}$ ($\mathcal{T}$ is set as $0.8$ in our current study). Overall, we report that from the employed cache side channel trace with 754139 data points, we successfully pinpoint a set of 31 data points that primarily contribute to the private image reconstruction (see Fig. 28 and Fig. 29).

Since each side channel data point is converted from a memory address (see Table 1), our retained "informative" side channel data points can thus be mapped back to certain functions in `libjpeg`. That is, we indeed use informative side channel records to isolate functions in `libjpeg` that potentially leak privacy. Fig. 27 depicts how this backward mapping and isolation are conducted. For instance, given a side channel record 0x55dba1628e62 marked as informative and notably contributes to the gradient, we use the address of the corresponding memory access instruction, 0x7f38daafd6b5, to isolate function `jsimd_convsamp_float`. That is, certain input-dependent memory access in `jsimd_convsamp_float` induces this cache line access and eventually contributes to the reconstruction of the user private input.

Fig. 28 reports a part of the logged side channel trace and marks several side channel data points in red which largely affects the gradient. We show their corresponding functions in `libjpeg` in Fig. 29. Overall, we report that this evaluation successfully pinpoints multiple critical functions in the `libjpeg` software that has input-dependent cache accesses. In particular, we note that this evaluation helps to "re-discover" some functions that have been reported by previous research (mostly

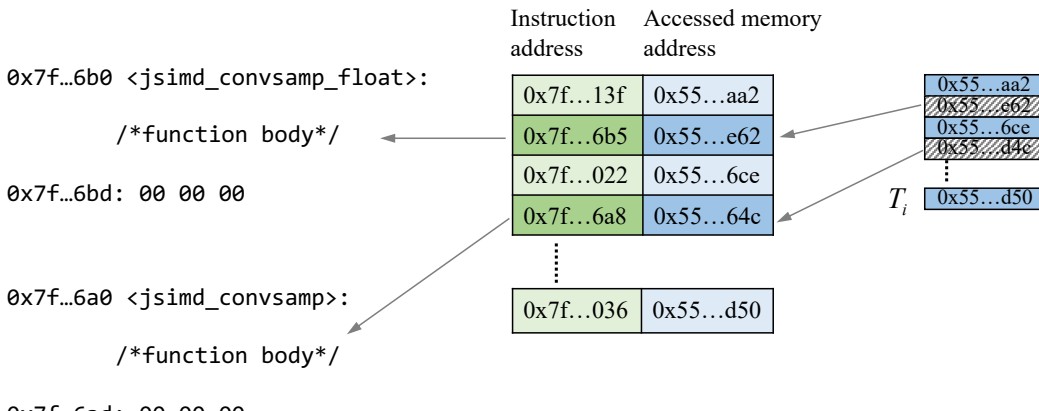

Figure 27: Mapping "informative" side channel data points back to functions in libjpeg.

with manual effort) as vulnerable to SCA: e.g., functions write_ppm, put_rgb and output_ppm which dump the decompressed raw image to the disk.

More importantly, this evaluation helps to pinpoint **new** functions that contribute to the private image reconstruction (and hence become vulnerabilities to SCA), such as jsimd_can_fdct_islow, jsimd_can_fdct_ifast, jsimd_convsamp and jsimd_convsamp_float. These functions primarily conduct image discrete cosine transformation (DCT) and decompression. We interpret this as a highly encouraging finding, in particular:

- As reviewed in Sec. 2, previous research uses manual effort (Xu et al., 2015; Hähnel et al., 2017) or formal methods (Doychev et al., 2013; Wang et al., 2017) to pinpoint program components that depend on inputs, which are program-specific and error-prone with low scalability.

- This research and our study in this section actually reveals a procedure where we leverage gradient to directly highlight which part of the logged side channel trace contributes to the synthesis of outputs. Then, we map the highlighted trace back to where they are derived from in the victim software to isolate vulnerable components (i.e., a *bug detection* tool).

We view it as a promising finding: our approach depicted in this section is general and can be launched fully automatically without requiring manual efforts or formal methods which are usually not scalable. As shown in this section, our tentative study not only re-discovers vulnerabilities that were found by previous research, but helps to identify, to our best knowledge, unknown program components that are vulnerable to SCA. Looking ahead, we would like to explore this direction as a follow-up of the present work.

```
...          ...          ...          0x55dba16169f4 ...
0x55dba1628acc 0x55dba1628ac9 0x7ffd1d241f18 0x55dba1628d60 0x55dba1628d68 0x55dba16286d4 0x55dba1620dd8 0x55dba162c6dd 0x55dba1616bf4
0x55dba1617177 0x55dba161717e 0x7ffd1d241f18 0x55dba1628729 0x55dba16286e8 0x55dba1620bdf 0x55dba1616b17 0x55dba1616a1d 0x55dba162b958
0x55dba1628acc 0x55dba1628ac 0x55dba162871b 0x55dba161be52 0x55dba161ee26 0x55dba1620bdc 0x55dba162c697 0x7f38dab5e6a0
0x55dba1617171 0x55dba161717a 0x55dba162cc40 0x55dba16297a0 0x55dba16297b0 0x55dba1616b98 0x55dba16169d7 0x55dba16286e3 0x55dba1616c02
0x55dba1628acc 0x55dba1628ac 0x55dba1628726 0x55dba1629e80 0x55dba1629ec8 0x55dba162c618 0x55dba1628d1c 0x55dba1620ef6 0x55dba162ba30
0x55dba1617176 0x55dba161717d 0x55dba162cc47 0x55dba161be26 0x55dba16286c6 0x55dba1616a58 0x55dba16286e4 0x55dba16297a8 0x7f38dab5e6a4
0x55dba1628acc 0x55dba1628ac3 0x55dba162871e 0x55dba161be51 0x55dba161ee25 0x55dba1628d1c 0x55dba1620dde 0x55dba1629eb8 0x55dba1616c10
0x55dba1617172 0x55dba161717b 0x55dba162cc41 0x55dba16290d0 0x55dba16290f4 0x55dba16286e4 0x55dba1629718 0x55dba16286c3 0x55dba162b928
0x55dba1628acc 0x55dba1628ac4 0x55dba1628727 0x55dba16286f0 0x55dba16286bc 0x55dba1620be2 0x55dba1629ed0 0x55dba1620ef5 0x7f38dab5e6a8
0x55dba1617175 0x55dba161717c 0x55dba162cc46 0x55dba161be53 0x55dba161ee27 0x55dba1629718 0x55dba16286e9 0x55dba16290ec 0x55dba1616c20
0x55dba1628acc 0x7ffd1d241f10 0x55dba1628723 0x55dba161be50 0x55dba161ee24 0x55dba1629ed0 0x55dba1620ddd 0x55dba16286b4 0x55dba162b8a8
0x55dba1617173 0x7ffd1d241f10 0x55dba162cc42 0x55dba1616b35 ...            0x55dba16286e9 0x55dba16290f8 0x55dba1620ef7 0x7f38dab5e6ac
0x55dba1628acc 0x7ffd1d241f28 0x55dba1628728 0x55dba162ccb5 0x55dba161ef2e 0x55dba1620be1 0x55dba16286d3 0x55dba1620ef4 0x55dba1616c12
0x55dba1617174 0x7ffd1d241f84 0x55dba162cc45 0x55dba16169f5 0x55dba16297c8 0x55dba16290f8 0x55dba1620ddf 0x55dba1616b5e 0x55dba162b8b4
0x7ffd1d241f10 0x7ffd1d241f10 0x55dba1628727 0x55dba1629e88 0x55dba1629e88 0x55dba16286d3 0x55dba1620ddc 0x55dba162c6de 0x7f38dab5e6b0
0x7ffd1d241f10 0x7ffd1d241f88 0x55dba162cc43 0x55dba1628725 0x55dba1628713 0x55dba1620be3 0x55dba1616b18 0x55dba1616a1e 0x55dba1616c04
0x7ffd1d241f28 0x7ffd1d241f18 0x55dba1628728 0x55dba161be56 0x55dba161ef2d 0x55dba1620be0 0x55dba162c698 0x55dba1628d64 0x55dba162b91c
0x7ffd1d241f84 0x7ffd1d241f90 0x55dba162cc44 0x55dba16297a0 0x55dba16290d4 0x55dba1616b99 0x55dba16169d8 0x55dba16286f2 0x7f38dab5e6b4
0x7ffd1d241f10 0x7ffd1d241f9c 0x7ffd1d241f10 0x55dba1629e88 0x55dba16286f9 0x55dba162c619 0x55dba1628d1c 0x55dba1620efa 0x55dba1616bf6
0x7ffd1d241f88 0x7ffd1d241f94 0x55dba162870a 0x55dba161ef2f 0x55dba1616a59 0x55dba16286e4 0x55dba16297a8 0x55dba162bc58
0x7ffd1d241f98 0x7ffd1d241f8c 0x7ffd1d241f28 0x55dba161be55 0x55dba161ef2c 0x55dba1628d1c 0x55dba1620de2 0x55dba1629eb0 0x7f38dab5e6c8
0x7ffd1d241f90 0x7ffd1d241f20 0x7ffd1d241fe4 0x55dba16290d4 0x55dba1616b6c ...            0x55dba1629718 0x55dba16286d3 0x55dba1616c14
0x7ffd1d241f9c 0x55dba162ad98 0x7ffd1d241f10 0x55dba16286ed 0x55dba162c6ec 0x55dba1620c3a 0x55dba1629ec8 0x55dba1620ef9 0x55dba162b940
0x7ffd1d241f94 0x7ffd1d241f80 ...            0x55dba161be57 0x55dba1616a2c 0x55dba16297c8 0x55dba16286e9 0x55dba16290e8 0x7f38dab5e6cc
0x7ffd1d241f8c 0x7ffd1d241f18 0x55dba1628733 0x55dba161be54 0x55dba1628d70 0x55dba1629e68 0x55dba1620de1 0x55dba16286c1 0x55dba1616c22
0x7ffd1d241f20 0x7ffd1d241f18 0x55dba162cccd 0x55dba1616b36 0x55dba1628739 0x55dba1628702 0x55dba16290f4 0x55dba1620efb 0x55dba162b97c
0x55dba162ad98 0x55dba1628acb 0x55dba162872f 0x55dba162ccb6 0x55dba161ef32 0x55dba1620c39 0x55dba16286d2 0x55dba1620ef8 0x7f38dab5e6d0)
0x7ffd1d241f80 0x55dba16171b8 0x55dba162cccb 0x55dba16169f6 0x55dba16297c0 0x55dba16290c4 0x55dba1620de3 0x55dba1616b5f 0x55dba1616c30
0x7ffd1d241f18 0x55dba1628ace 0x55dba1628730 0x55dba1629e88 0x55dba16286e0 0x55dba1616be0 0x55dba162c6df 0x55dba162bb54
0x7ffd1d241f18 0x55dba16171bf 0x55dba162cccc 0x55dba1628733 0x55dba1628715 0x55dba1620c3b 0x55dba1616b19 0x55dba1616a1f 0x7ffd1d241fe0
0x55dba1628ad1 0x55dba1628aca 0x7ffd1d241f10 0x55dba161be5a 0x55dba161ef31 0x55dba1620c38 0x55dba162c699 0x55dba1628d68 0x7ffd1d241fd8
0x55dba16171b0 0x55dba16171b9 0x7ffd1d241f10 0x55dba16297a8 0x55dba16290d4 0x55dba1616aaf 0x55dba16169d9 0x55dba1628704 0x55dba1615878
0x55dba1628acb 0x55dba1628acd 0x7ffd1d241f28 0x55dba1629e88 0x55dba16286fb 0x55dba162c62f 0x55dba1628d1c 0x55dba1620efe 0x7ffd1d241fd0
0x55dba16171b7 0x55dba16171be 0x7ffd1d242004 0x55dba1628715 0x55dba161ef33 0x55dba1616f6f 0x55dba16286e3 0x55dba16297b0 0x55dba162a9e0
0x55dba1628ad0 0x55dba1628ac9 0x7ffd1d241f10 0x55dba161be59 0x55dba161ef30 0x55dba1628d78 0x55dba1620de6 0x55dba1629ea8 0x7ffd1d241fc0
0x55dba16171b1 0x55dba16171ba ...            0x55dba16290d4 0x55dba1616b6d 0x55dba162872f 0x55dba1629718 0x55dba16286e3 0x55dba162a8a8
0x55dba1628acb 0x55dba1628acb 0x55dba1628706 0x55dba16286f9 0x55dba162c6ed 0x55dba1620c3e 0x55dba1629ec8 0x55dba1620efd 0x55dba162a8f8
0x55dba16171b6 0x55dba16171bd 0x55dba161be41 0x55dba161be5b 0x55dba1616a2d 0x55dba16297d0 0x55dba16286e8 0x55dba16290e4 0x7ffd1d241fb8
0x55dba1628acf 0x55dba1628ac9 0x55dba16290cc 0x55dba161be58 0x55dba1628d6c 0x55dba1629e68 0x55dba1620de5 0x55dba16286d0 0x7ffd1d241fb0
0x55dba16171b2 0x55dba16171bb 0x55dba1628ae4 0x55dba1616b37 0x55dba1628739 ...            0x55dba16290f4 0x55dba1620eff 0x7ffd1d241fa8
0x55dba1628acc 0x55dba1628aca 0x55dba161be43 0x55dba162ccb7 0x55dba161ef36 0x55dba162869c 0x55dba16286d1 0x55dba1620efc 0x7ffd1d241fa0
0x55dba16171b5 0x55dba16171bc 0x55dba161be40 0x55dba16169f7 0x55dba16297b8 0x55dba1620d47 0x55dba1620de7 0x55dba1616b60 0x7ffd1d241f98
0x55dba1628ace 0x7ffd1d241f10 0x55dba1616b31 0x55dba1628d64 0x55dba1629e90 0x55dba1620d44 0x55dba16169d8 0x55dba162c6e0 0x7ffd1d241f90
0x55dba16171b3 0x7ffd1d241f10 0x55dba162ccb1 0x55dba162874b 0x55dba1628717 0x55dba1616af2 0x55dba1616b1a 0x55dba1616a20 0x7ffd1d241f88
0x55dba1628acd 0x7ffd1d241f28 0x55dba16169f1 0x55dba161be5e 0x55dba161ef35 0x55dba162c672 0x55dba162c69a 0x55dba1628d6c 0x7ffd1d242020
0x55dba16171b4 0x7ffd1d241fa4 0x55dba1628d5c 0x55dba16297a8 0x55dba16290d8 0x55dba1616b2f 0x55dba16169b3 0x55dba16169db 0x55dba162871a 0x7ffd1d242000
0x7ffd1d241f10 0x7ffd1d241f10 0x55dba162872b 0x55dba1629e80 0x55dba16286fe 0x55dba1628d34 0x55dba1628d18 0x55dba1620f02 0x7ffd1d241f78
0x7ffd1d241f10 0x7ffd1d241fa8 0x55dba161be46 0x55dba162872e 0x55dba161ef37 0x55dba16286a2 0x55dba16286e0 0x55dba16297b8 0x7ffd1d242008
0x7ffd1d241f28 0x7ffd1d241f28 0x55dba1629798 0x55dba161be5d 0x55dba161ef34 0x55dba1620d4a 0x55dba1620dea 0x55dba1629e98 0x55dba16158b4
0x7ffd1d241fa4 0x7ffd1d241fb0 0x55dba1629e78 0x55dba16290d0 0x55dba1616b6e 0x55dba1629748 0x55dba1629710 0x55dba16286f7 0x55dba1617fef
0x7ffd1d241f10 0x7ffd1d241fbc 0x55dba1628713 0x55dba1628710 0x55dba162c6ee 0x55dba1629f30 0x55dba1629ec8 0x55dba1620f01 0x55dba1617ff0
0x7ffd1d241fa8 0x7ffd1d241fb4 0x55dba161be45 0x55dba161be5f ...            0x55dba1628697 0x55dba16286e8 0x55dba16290dc 0x55dba1617ff1
0x7ffd1d241fb8 0x7ffd1d241fac 0x55dba16290cc 0x55dba161be5c 0x55dba162d075 0x55dba1620d49 0x55dba1620de9 0x55dba16286e1 0x55dba1617ff2
0x7ffd1d241fb0 0x7ffd1d241f20 0x55dba16286f1 0x55dba1616b38 0x55dba1616b48 0x55dba1629128 0x55dba16290f4 0x55dba1620f03 0x55dba1617ff3
0x7ffd1d241fbc 0x55dba162ada0 0x55dba161be47 0x55dba162ccb8 0x55dba1616b49 0x55dba162869f 0x55dba16286d0 0x55dba1620f00 0x55dba1617ff4
0x7ffd1d241fb4 0x7ffd1d241fa0 0x55dba161be44 0x55dba16169f8 0x55dba162cfb6 0x55dba1620d4b 0x55dba1620deb 0x55dba1616b61 0x55dba1617ff5
0x7ffd1d241fac 0x7ffd1d241f18 0x55dba162ccb2 0x55dba1628d64 0x55dba162d076 0x55dba1620d48 0x55dba1620de8 0x55dba162c6e1 0x7ffd1d242008
0x7ffd1d241f20 0x7ffd1d241f18 0x55dba162ccb2 0x55dba1628735 0x55dba1616b4a 0x55dba1616af3 0x55dba1616b1b 0x55dba1616a21 0x7ffd1d242010
0x55dba162ada0 0x55dba1628acf 0x55dba16169f2 0x55dba161be62 0x55dba1616b4b 0x55dba162c673 0x55dba162c69b 0x55dba1628d70 0x7ffd1d242018
0x7ffd1d241fa0 0x55dba16171f8 0x55dba1628d60 0x55dba16297a8 0x55dba162cfb7 0x55dba16169b3 0x55dba16169db 0x55dba1628733 0x7ffd1d242000
0x7ffd1d241f18 0x55dba1628ad2 0x55dba1628732 0x55dba1629e78 0x55dba162d077 0x55dba1628d34 0x55dba1628d18 0x55dba1620f06 0x7ffd1d241f88
0x7ffd1d241f18 0x55dba16171ff 0x55dba161be4a 0x55dba1628718 0x55dba1616b4c 0x55dba16286b4 0x55dba16286df 0x55dba16297c0 0x7ffd1d241f90
0x55dba1628ad5 0x55dba1628acf 0x55dba1629a7a0 0x55dba1616b4d 0x55dba161be61 0x55dba1616b4d 0x55dba1620d4e 0x55dba1620dee 0x55dba1629e90 0x7ffd1d241f98
0x55dba16171f0 0x55dba16171f9 0x55dba1629e78 0x55dba16290cc 0x55dba162cfb8 0x55dba1629748 0x55dba1629710 0x55dba162870f 0x7ffd1d241fa0
0x55dba1628acc 0x55dba1628ad2 0x55dba1628718 ...            0x55dba162d078 0x55dba1629f30 0x55dba1629ec0 0x55dba1620f05 0x7ffd1d241fa8
0x55dba16171f7 0x55dba16171fe 0x55dba161be49 0x55dba162c6f7 0x55dba1616b4e 0x55dba16286a9 0x55dba16286e8 0x55dba16290d8 0x7ffd1d241fb0
0x55dba1628ad4 0x55dba1628acf 0x55dba16290cc 0x55dba162871b 0x55dba1616b4f 0x55dba1620d4d 0x55dba1620ded 0x55dba16286f7 0x7ffd1d241fb8
0x55dba16171f1 0x55dba16171fa 0x55dba16286f7 0x55dba162c6f1 0x55dba162cfb9 0x55dba1629128 0x55dba16290f0 0x55dba1620f07 0x7ffd1d242018
0x55dba1628acc 0x55dba1628ad1 0x55dba161be4b 0x55dba1628aa2 0x55dba162d079 0x55dba16286b1 0x55dba16286cd 0x55dba1620f04 0x7ffd1d242010
0x55dba16171f6 0x55dba16171fd 0x55dba161be48 0x55dba162c6f6 0x55dba1616b50 0x55dba1620d4f 0x55dba1620def 0x55dba1616b62 0x55dba162b654
0x55dba1628ad3 0x55dba1628acf 0x55dba1616b33 0x55dba1628711 0x55dba1616b51 0x55dba1620d4c 0x55dba1620dec 0x55dba162c6e2 0x7ffd1d241fc0
0x55dba16171f2 0x55dba16171fb 0x55dba162ccb3 0x55dba162c6f2 0x55dba1616af4 0x55dba1616b1c 0x55dba1616a22 0x55dba162a944
0x55dba1628ace 0x55dba1628ad0 0x55dba16169f3 0x55dba1628abc 0x55dba162d07a 0x55dba162c674 0x55dba162c69c 0x55dba1628d70 0x7ffd1d241fd8
0x55dba16171f5 0x55dba16171fc 0x55dba1628d60 0x55dba162c6f5 0x55dba1616b4b 0x55dba16169dc 0x55dba16169dc 0x55dba1628740 0x55dba1615880
0x55dba1628ad1 0x7ffd1d241f10 0x55dba162872f 0x55dba16286f 0x55dba1616b53 0x55dba1628d34 0x55dba1628d18 0x55dba1620f0a 0x7ffd1d242030
0x55dba16171f3 0x7ffd1d241f10 0x55dba161be4e 0x55dba162c6f3 0x55dba162cfbb 0x55dba162869c 0x55dba16286dd 0x55dba16297c0 0x7ffd1d242030
0x55dba1628acf 0x7ffd1d241f10 0x55dba16297a0 0x55dba1628ad6 0x55dba162d07b 0x55dba1620d52 0x55dba1620df2 0x55dba1629e90 0x55dba162a96c
0x55dba16171f4 0x7ffd1d241fc4 0x55dba1629e80 0x55dba162c6f4 0x55dba1616b54 0x55dba1629748 0x55dba1629710 0x55dba162871c 0x55dba1616c60
0x7ffd1d241f10 0x7ffd1d241f10 0x55dba1628714 0x7ffd1d241f10 0x55dba1616b55 0x55dba1629f30 0x55dba1629ec8 0x55dba1620f09 0x7ffd1d241fe0
0x7ffd1d241f10 0x7ffd1d241f10 0x55dba161be4d 0x7ffd1d241f10 0x55dba162cfbc 0x55dba16286b1 0x55dba16286e5 0x55dba16290d8 0x55dba162bac0
0x7ffd1d241f280 0x7ffd1d241fd8 0x55dba16290d0 0x7ffd1d241f28 0x55dba162d07c 0x55dba1620d51 0x55dba1620df1 0x55dba1628704 0x7f38dab5e684
x7ffd1d241fc4 0x7ffd1d241fd0 0x55dba16286f6 0x7ffd1d242004 0x55dba1616b56 0x55dba1629128 0x55dba16290f4 0x55dba1620f0b 0x55dba1616c62
0x7ffd1d241f10 0x7ffd1d241fdc 0x55dba161be4f 0x7ffd1d241f10 0x55dba1616b57 0x55dba16169b3 0x55dba16286cd 0x55dba1620f08 0x55dba162bb90
0x7ffd1d241fc8 0x7ffd1d241fd4 0x55dba161be4c 0x7ffd1d242008 0x55dba162cfbd 0x55dba1620d53 0x55dba1620df3 0x55dba1616b63 0x7f38dab5e688
0x7ffd1d241fd8 0x7ffd1d241fcc 0x55dba1616b34 0x7ffd1d242018 0x55dba162d07d 0x55dba1620d50 0x55dba1620df0 0x55dba162c6e3 0x55dba1616c70
...          ...          0x55dba162ccb4 ...          ...          ...          ...          ...          ...
```

Figure 28: Part of the logged side channel trace and informative side channel data points marked in red.

| Function | Instruction Address | Accessed Memory Address |
|---|---|---|
| put_rgb | 0x7f38dab131b8 | 0x55dba1628acc |
| put_rgb | 0x7f38dab131f9 | 0x55dba1628acc |
| put_rgb | 0x7f38dab131b8 | 0x55dba1628ac4 |
| put_rgb | 0x7f38dab131b8 | 0x55dba1628ac6 |
| put_rgb | 0x7f38dab13109 | 0x7ffd1d241f18 |
| put_rgb | 0x7f38dab131fe | 0x55dba162cc44 |
| put_demapped_rgb | 0x7f38dab13010 | 0x7ffd1d241fe4 |
| put_rgb | 0x7f38dab131e0 | 0x55dba162cccb |
| jsimd_can_fdct_islow | 0x7f38daafd6cd | 0x55dba161be41 |
| jsimd_convsamp | 0x7f38daafd6a8 | 0x55dba1628d5c |
| jsimd_convsamp | 0x7f38daafd6a8 | 0x55dba1628d60 |
| jsimd_convsamp_float | 0x7f38daafd6bc | 0x55dba1629e80 |
| jsimd_convsamp | 0x7f38daafd6a8 | 0x55dba1628d60 |
| jsimd_convsamp_float | 0x7f38daafd6bc | 0x55dba1629e88 |
| jsimd_convsamp_float | 0x7f38daafd6b5 | 0x55dba161be62 |
| jsimd_can_fdct_islow | 0x7f38daafd6cd | 0x55dba161be61 |
| put_rgb | 0x7f38dab131db | 0x55dba16286f5 |
| jsimd_can_fdct_islow | 0x7f38daafd6cd | 0x55dba161ee25 |
| jsimd_can_fdct_islow | 0x7f38daafd6cd | 0x55dba161ef2d |
| jsimd_can_fdct_islow | 0x7f38daafd6cd | 0x55dba161ef35 |
| output_ppm | 0x7f38dab10293 | 0x55dba162cfb7 |
| output_ppm | 0x7f38dab10293 | 0x55dba162cfb9 |
| jsimd_convsamp_float | 0x7f38dab102b0 | 0x55dba1616b50 |
| jsimd_convsamp_float | 0x7f38daafd6b5 | 0x55dba1620be2 |
| jsimd_can_fdct_islow | 0x7f38daafd6cd | 0x55dba1620c39 |
| jsimd_convsamp | 0x7f38daafd6a8 | 0x55dba1628d34 |
| jsimd_convsamp_float | 0x7f38daafd6bc | 0x55dba1629f30 |
| jsimd_can_fdct_islow | 0x7f38daafd6cd | 0x55dba1620de1 |
| jsimd_convsamp_float | 0x7f38daafd6b5 | 0x55dba1620de6 |
| jsimd_can_fdct_islow | 0x7f38daafd6cd | 0x55dba1620efd |
| jinit_write_ppm | 0x7f38dab03f1d | 0x55dba1616c14 |

Figure 29: Isolated functions in libjpeg that contribute to the reconstruction of user private images.

