# OpenReview forum: "Private Image Reconstruction from System Side Channels Using Generative Models"
_ICLR.cc/2021/Conference — ICLR 2021 Poster_

### Official Review · AnonReviewer1 · 2020-10-21
**Review of Private Image Reconstruction from System Side Channels Using Generative Models**

**Rating:** 8
**Confidence:** 3

**Review:**

Objective of the paper:
The objective of the paper is to show that the side channel information (based on timing, etc.) for image analysis software can be used to reconstruct reasonably accurate to highly accurate approximations of user images.  While this was previously known, in this paper, the authors use machine learning techniques to more efficiently "learn" how to use this side-channel information, developing generative models that allow image reconstruction.

Strong Points:
1)  The use of machine learning techniques as an "attack methodology" for making use of side channel information seems to be a potentially powerful methodology.
2)  The results shown here seem quite strong.
3)  The application (image analysis) is well chosen;  the side channels under consideration seems quite plausible.
4)  The code is made available.

Weak Points:
1)   There are obvious counters to this approach;  once one knows that this type of software is being targeted, one can introduce various randomization (for memory accesses, timing, etc.) to lessen the power of side channel attacks.  That is, this may only be a reasonable attack because nobody has to this point considered it important to avoid side channel information being released by image analysis software.
2)   It's a little unclear (to this reviewer) how much "information" the learning algorithm is seeded with.  That is, for these image sets, how similar/distinct are the underlying images?  Your examples seem to capture a lot of the original image, but it's not clear to me how to interpret what the algorithm is starting with.  (Essentially, would these results scale to much larger sets of images?  How much does working with the limited set of celebrity faces already help yield the strong results?  This is tackled somewhat in Appendix F, so the authors do seem aware of this point.)
3)  It seems like you're considering all the different side channels separately?  Is there any way to combine the multiple side channels (into a mega-side-channel?) and do better?  (I apologize if I'm missing something -- I'd expect to see a result set with each side information, as you've given, but then results with all combined.)
4)  This reviewer does not have good insight into why this type of side information would be useful in determining how to reconstruct images in the manner done here.  Some description or intuition as to how accessed cache lines could intuitively be used to determine an approximation to the original picture would be welcome.  (Understood you have limited space.)
5)   The paper does seem like a first step and there remains significant room for improvement (improved side channels, improved scalability, other applications).

Overall Rating:
Despite the (relatively minor) weak points, the paper seems to be a strong proof of concept that machine learning can greatly increase the risk of side channel attacks, because they can essentially automate the task of determining how to use the side channel information to obtain the desired result.  This seems to be an important message for the community, in particular as I expect this framework is not specific to image analysis software, but can be applied more generally.

Questions for Authors:
I would very much like to see some sort of experiment where noise is injected in the side channel information.  One can view this as a setting where one obtains limited side channel information for some reason, or as a setting where the program is modified in some way to lessen the amount/value of side information.

Similarly, as discussed above, is there some intuition as to why the side information yields such reasonable results?  (Again, and how this relates to the limited data sets?)  And whether or not you could combine multiple types of side information naturally using this approach?

Other Feedback:
Overall thee paper seems well written, and you do provide a great deal of data.

---

> ### Author Response · Authors · 2020-11-15
> **Response to Reviewer #1**
>
>
> Thank you for your efforts reviewing our paper and providing helpful comments and suggestions. We hope you don't mind our long response, and looking forward to hearing your response.
>
> **There are obvious counters to this approach;**
>
> Thank you for the comments. Overall, we have seen a number of papers that propose methods to mitigate SCA from either software aspect (e.g., designing “secret-independent” cache access software) or system aspect. However, one key concern is the incurred extra execution slowdown. We will extend the paper to review potential mitigation methods of our SCA.
>
> **That is, for these image sets, how similar/distinct are the underlying images?**
>
> Thank you for the question. We were also curious about that and therefore besides some “specific” datasets (e.g., Face and human gestures), we also evaluated a general dataset, mini-Imagenet. Recovering images from this dataset denotes scenarios where attackers have zero prior-knowledge. As reported in Table 3 of the paper (page 8), the attack success rates, on average, are 4.0 times higher than the baseline (random guess).
>
> **It seems like you're considering all the different side channels separately? Is there any way to combine the multiple side channels (into a mega-side-channel?) and do better?**
>
> Yes. We consider all different side channels separately.
>
> We believe combining multiple side channels is technically feasible. On the other hand, it seems not aligned with how typical SCA is launched in the real-world scenarios. Overall, we have evaluated three side channels in the paper:
> - Cache based side channel: denoting a commonly used attack scenario where attackers share the same cache unit with the victim software.
> - Page table side channel: denoting another common scenario where attackers can monitor the OS page table access pattern.
> - Read/write side channel: this side channel was considered to be less informative. However, we, *for the first time*, show that quality images can be synthesized from this side channel. Such findings reveal the strength of our approach.
>
> Typically, if an attacker can only use page table side channels, it indicates that he is *not* capable of using cache line-based side channel which is indeed finer-grained and more informative.
>
> As shown in Table 1 of our paper, cache based side channel indeed “subsumes” page table and read/write side channels. Evaluation consistently shows that image recovered by cache-based side channels are better than images recovered from the other two side channels.
>
> Overall, evaluating the composition of side channels seems not consistent with assumptions of real-world attacks. We will extend the paper to further clarify this point and avoid any over-claim or confusions.
>
> **Some description or intuition as to how accessed cache lines could intuitively be used to determine an approximation to the original picture would be welcome.**
>
> Thank you for the suggestion. For the past few days, we have launched experiments to isolate which part of the side channel trace contributes to the construction of pictures. We then map the critical part of the side channel trace back to particular functions in the `libjpeg` software. This way, we isolate certain functions in the `libjpeg` that are vulnerable and primarily contribute to image reconstruction.
>
> The findings are highly encouraging and inspiring. **please refer to our response to `what information in the side channel is used for image reconstruction` to Reviewer #2**.
>
> **I would very much like to see some sort of experiment where noise is injected in the side channel information.**
>
> Thank you for the suggestions. For the past few days, we have launched experiences to add noise into the collected side channel traces. The overall results are promising, indicating the resilience of our framework to noise. **Please refer to our response to `Is it robust to noise or shifting?` to Reviewer #2.**

---

### Official Review · AnonReviewer3 · 2020-10-28
**The authors present a framework to reconstruct private user images using a GAN trained on side channel information obtained from image libraries accessing the image.**

**Rating:** 7
**Confidence:** 4

**Review:**


#### Summary:
The authors present a framework that uses a combination of   VAE and GAN to recover private user images using Side channel analysis of memory access . A VAE-LP model first reconstructs a coarse image from side channel information which is reshaped and processed using a convolutional network. The output of the VAE-LP model is refined using a GAN to add fine details. Compelling results are demonstrated for recovery of private information and state of art metrics are reported.

#### Strengths:
1. The paper is well written with great attention to detail. All design decisions are well motivated and have been adequately explained.
2. State of the art qualitative results are demonstrated for recovery of images using SCA .
3. The paper is an extremely interesting application of the VAE-GAN framework.
4. The proposed solution offers a way to recover private information without requiring specialized knowledge of the system architecture, which is a great advantage as it is potentially generalizable to a wide variety of architectures.
5. To the best of the reviewer's knowledge, the treatment of related work is adequate and the positioning of the current work in the context of prior art is made very clear.
6. Analysis is performed on a wide variety of datasets indicating that the solution is not over-engineered to a particular library or architecture.
7. The appendix presents a lot of interesting experiments and analysis. Particularly, classification performance and recovery quality on images trained using a more generic rather than domain specific dataset.

#### Weaknesses:
1. Although the authors mention that direct comparisons are hard to provide to previous methods due to lack of public implementation. Few qualitative examples of recovered images from prior art (relegated to the appendix) would help motivate the need for the proposed approach better.
2. It is not clear why the L1 loss is necessary for the second stage in section 3.3 . Particularly, as opposed to a more well studied framework like Pix2pix[1], which imposes a L1 loss at the output with a ground truth image. Because at training time we know the GT image that we are recovering. An ablation study demonstrating the quality without this L1 loss would be insightful.
3. Reporting image level metrics is instructive in understanding how far the reconstructed image is from the ground truth image. Particularly, SSIM and LPIPS[2] score of generated image w.r.t ground truth image.
4. Although the formulation is interesting. The novelty is somewhat limited as it is straightforward application of the VAE-GAN framework to SCA data.

#### Questions:
1. The exact nature of the trace encoding is unclear. Are memory locations fed directly into the network? If so how are the normalized, because assuming 16 bit addresses, the range of values occupied
2. The authors mention that libraries like libjpeg generates 700k side channel "signals"  in Section 1. What is the nature of these signals? Particularly are the single bits of information/ text/ memory locations?
3. Although beyond the scope of the presented work, this framework allows for access of private information. Such methods raise obvious ethical concerns on part of services which offer ML solution. A small statement about how this form of attack can be protected against would be instructive in strengthening the narrative.
4. The motivation to use a CNN to process trace matrix is unclear. Particularly, are assumptions of equivariance made on the access patterns?  Apart from the architectural convenience of weight sharing is there a stronger motivation to assume a convolutional structure on the input? An MLP(or 1*1 conv) with pooling run on the same $$N \times N \times K$$ encoded trace matrix would provided some insights regarding the need for a CNN to process the trace information.
5. It is not clear what the generated *Dec* refers to in Sec. 3.2.

#### Minor:
subsumed -> assumed ? in section 3.2

#### Justification of rating:
Although the authors present an interesting application of the VAE-GAN framework for recovery of private information from SCA, the formulation of the problem is remarkably similar to conditional-VAE. It would be interesting to further analyze the learnt representation from the VAE-LP to see if certain "code-blocks" or functions can be isolated. Having said that, the paper is very well written and thorough in its analysis. The reviewer also believes that a systems/privacy audience would be able to appreciate the contributions of this work better.

#### Updates
In light of the responses provided by the authors, and the thorough nature of the additional experiments I am inclined to increase the ratings of this paper. The followup responses answer most concerns raised by reviewers and provide supporting evidence. However, the reviewer maintains that this work could potentially be of more interest to a privacy audience who may better be able to appreciate the use case considered in the presented work.

[1] Isola, Phillip, et al. "Image-to-image translation with conditional adversarial networks." Proceedings of the IEEE conference on computer vision and pattern recognition. 2017.
[2] Zhang, Richard, et al. "The unreasonable effectiveness of deep features as a perceptual metric." Proceedings of the IEEE conference on computer vision and pattern recognition. 2018.

---

> ### Author Response · Authors · 2020-11-15
> **Response to Reviewer #3**
>
>
> Thank you for your efforts reviewing our paper and providing helpful comments and suggestions. We hope you don't mind our long response, and looking forward to hearing your response.
>
> **Few qualitative examples of recovered images from prior art.**
>
> Thank you for the suggestions. We note that it is really difficult, if at all possible, to somehow re-run previous attacks for comparison. As discussed in the paper, previous attacks ([1,2]]) seem to use domain-specific knowledge, expertise, and even manual efforts to “distill” the collected side channel traces and map then back to secrets. In contrast, we, for the first time, explore the *synthesis* capability of generative models in this field. From a holistic perspective, secrets are reconstructed by **mapping** & **complementing**, which is quite different from previous works.
>
> We have released all artifacts (code; data) of this research for results verification. We will definitely maintain our codebase to facilitate comparison of future research.
>
> [1] Controlled-channel attacks: Deterministic side channels for untrusted operating systems. IEEE Security & Privacy 2015
>
> [2] High-resolution side channels for untrusted operating systems. USENIX ATC 2017
>
> **why the L1 loss is necessary for the second stage in section 3.3**
>
> Thank you for the question. L1 loss performs better on capturing the low-frequency part of an image, which is also acknowledged in Pix2pix [1].
>
> Overall, We show that L1 loss, as a common setting, is already sufficient to conduct SCA and recover user inputs of high quality. Further research on potentially improving model structures can take this present work and its setting as “baseline” for comparison.
>
> [1] Image-to-Image Translation with Conditional Adversarial Networks. CVPR 2017
>
> **Reporting image level metrics is instructive in understanding how far the reconstructed image is from the ground truth image.**
>
> Thank you for the suggestion. For the past few days, we used LPIPS to evaluate the synthesized images. We now report the quantitative results as follows:
>
> | | K=1,N=100 | K=5,N=100 | K=20,N=100 |
> | :--:| :--: | :--: | :--: |
> | VAE-LP  | 23.02% | 47.92% | 72.40% |
> | VAE-LP&GAN  | 39.64% | 63.12% | 81.42% |
>
> Compared with our results reported in the paper, the accuracy of GAN output is reduced by 10% and reduced even more on output of VAE-LP. It’s reasonable since we did NOT train the model with perceptual loss.
>
> Overall, we wish to clarify that the purpose of SCA is to precisely reconstruct the ground truth image. We pick the current metric given that it sufficiently facilitates reconstructing images that can be matched to the ground truth images. Our framework can serve as the baseline for further enhancement or fine-tuning in different scenarios or with different requirements.
>
> **straightforward application of VAE-GAN framework**
>
> We wish to clarify that the present research is not a simply straightforward application of VAE-GAN framework. We wish to clarify the novelty of this work from the following aspects:
>
> - From the perspective of the research idea, this present research, for the first time, apply generative models to infer user private images. Note that conventional secret reconstruction is primarily an “one-to-one” mapping procedure, while we, for the first time, explore the *synthesis* capability of generative models in this field. From a holistic perspective, secrets are reconstructed by **mapping** & **complement**.
> - From the method perspective, we addressed a number of domain-specific challenges and delivered a practical framework to exploit real-world software. In particular, to smoothly learn from side channel traces which are sparse (only a small portion is private-related), lengthy (each trace has over 700K data points), and have locality, we employ a CNN layer. The side channel traces can thus be integrated into the generative framework.
> - From the empirical perspective, our method has already recovered images with highly promising quantitative and qualitative results. We discussed the potential tradeoff of the current design (generalizability). Our work, with code already released, serves as the **baseline** for further research on model structure improvement.

---

> > ### Author Response · Authors · 2020-11-15
> > **Cont'd**
> >
> >
> > **The exact nature of the trace encoding is unclear. Are memory locations fed directly into the network? If so how are the normalized, because assuming 16 bit addresses, the range of values occupied**
> >
> > Thank you for the question. We use Intel Pin to log memory locations (“memory addresses”), and then convert each memory address into a corresponding side channel data point (following Table 1 in the paper where  `addr` is a “memory location”).  These side channel data points are fed into the network.
> >
> > We agree the address, which is 48-bit, has a large range of values. On the other hand, we normalize the memory address value, which are discrete integers, into continuous values within [0,1]. Overall, while 48-bit integers have a large range, we wish to clarify that side channel data points indeed vary within a small range. For instance, for cache based side channels, the possible values are limited by the total number of CPU cache units. In all, side channel data points are large values ranging in a relatively small range.
> >
> > It seems `the range of values occupied` in the question is somehow broken. Kindly advise us in case we misunderstood your question or you would like to see further clarifications.
> >
> > **side channel "signals"**
> >
> > We apologize for the confusion. Overall, side channel signals represent the side channel data points converted from memory addresses. It does not have a single bit of the raw data. As aforementioned, for each logged memory location whose address is `addr`, we follow Table 1 in the paper to convert `addr` into a side channel data point.
> >
> > We will extend the draft and clarify this point to avoid misleading.
> >
> > **ethical statement**
> >
> > Thank you very much for pointing this out. We will definitely add an ethical statement in the revision. We will also extend the paper on potential mitigations of our attack.
> >
> > **The motivation to use a CNN to process trace matrix is unclear**
> >
> > We wish to clarify that the side channel trace is sparse, lengthy, and has locality. We choose CNN because of its good performance on capturing the locality of the input data. We have no assumption of the access pattern or the convolutional structure of the inputs. An MLP(or 1*1 conv) with pooling,  we believe, can also work on our framework.
> >
> > ***Dec* refers to in Sec. 3.2.**
> >
> > *Dec* means Decoder, which is used for both image data flow and trace data flow. We will clarify this point in the paper to avoid confusions.
> >
> > **analyze the learnt representation from the VAE-LP to see if certain "code-blocks" or functions can be isolated.**
> >
> > Thank you very much for the suggestion! In the past few days we launched experiences to visualize which code-blocks can be isolated that contribute to information recovery. The results are quite interesting. **Please refer to our response to `what information in the side channel is used for image reconstruction` to Reviewer #2**.

---

### Official Review · AnonReviewer4 · 2020-10-28
**Interesting Applications but Not Interesting Method**

**Rating:** 5
**Confidence:** 3

**Review:**

The authors proposed a representation and generation model to reconstruct the image signals from system side channel signals. The task itself is interesting and novel, demonstrating the first efforts and impressive performance on recovering noisy side channel signals. The work will potentially inspire more attempts to conduct attack and security research on computer internal signals.

Pros:

+ A clear problem statement and description of the proposed method, and very impressive performance in both visualization and quantitative metrics.
+ A good analysis of the generalization ability of the model.

Cons:
- The method itself makes sense to me, but not novel. The authors propose to share latent codes between two modalities and train them jointly. And then use GAN to further refine the image quality for visualization. The overall pipeline is very common in image restoration and generation area.
- The 'learn the prior' method may fail to learn a good latent space without another prior distribution regularization. It's better to compare it with first training the VAE with image encoders using fixed gaussian prior, and then train the latent space jointly while fixing the image encoder. I believe the latent space will be better regularized and smooth. However, the tasks do not require you to generate image from the latent space, the advantages may not be revealed from the reconstruction results.
- The performance of the model trained on mini-ImageNet is poor, even though the authors state that it still reconstructs the basic structures. However, it also reminds us that the good reconstruction results are highly related to the small image training database domain. It may be harder for larger databases or other unseen data.
- Will the organization of the matrix influence the results? What if there are random zero paddings or other errors in the log?
- It's better to analyze and think about the possible solution to avoid these kinds of attacks. Otherwise, purely talking about the attacks strategies may raise some ethics concerns of the paper.

In summary, the task sounds interesting to me, but the overall method is not novel at all. The authors utilized a totally old technical model and training method in a brand new topic by formatting the input modalities. The results seem impressive in smaller dataset, but it still not convincing enough when testing the model on cross-domain data. Besides, there are possible ethics issues if no discussion related to avoiding the attacks is covered.

Update:
Thanks for all the authors' feedback and the make-up experiments as well as revisions. I appreciate the authors' efforts on solving the privacy problem and new input forms, but I really think the paper made no contributions to 'learning representation'.  The VAE and GAN-based model is not problem-specific or novelly designed for the task. Instead, it is a general framework for any reconstruction problems. I do believe this will make a strong submission to other conferences related to information. In this case, I will not change my rating.

---

> ### Author Response · Authors · 2020-11-15
> **Response to Reviewer #4**
>
>
> Thank you for your efforts reviewing our paper and providing helpful comments and suggestions. We hope you don't mind our long response, and looking forward to hearing your response.
>
> **Novelty:**
>
> Thank you for the comments. We wish to clarify the novelty of this work from the following aspects:
> - From the perspective of the research idea, this present research, for the first time, apply generative models to infer user private images. Note that conventional secret reconstruction is primarily an “one-to-one” mapping procedure, while we, for the first time, explore the *synthesis* capability of generative models in this field. From a holistic perspective, secrets are reconstructed by **mapping** & **complement**.
> - From the method perspective, we addressed a number of domain-specific challenges and delivered a practical framework to exploit real-world software. In particular, to smoothly learn from side channel traces which are sparse (only a small portion is private-related), lengthy (each trace has over 700K data points), and have locality, we employ a CNN layer. The side channel traces can thus be integrated into the generative framework.
> - From the empirical perspective, our method has already recovered images with highly promising quantitative and qualitative results. We discussed the potential tradeoff of the current design (generalizability). Our work, with code already released, serves as the **baseline** for further research on model structure improvement.
>
> **It may be harder for larger databases or other unseen data.**
>
> Thank you for the question. We indeed had similar concerns during this research. Therefore, we measured the “generalizability” of our method by using a more general dataset, mini-Imagenet. As reported in Table 3 of the paper (page 8), the attack success rates, on average, are 4.0 times higher than the baseline -- random guess.
>
> Overall, the images recovered when using mini-Imagenet as the training data are visually worse than images recovered using specialized datasets. We consider this evaluation faithfully reveals the full potential and *trade-off* of our research. This also helps to avoid any over-claim or misleading of the results.
>
> We also want to clarify that when training the model on mini-ImagNet, we put images from different classes together and **provide no information of the class label**, which is very challenging and is different from the existing methods of training generative models on this kind of dataset.
>
> **Will the organization of the matrix influence the results?**
>
> Thank you for the question. We currently use a straightforward setting by simply dividing a trace into K segments of equal length; N adjacent points in a segment are put into one row, in total N rows. This setting is independent from the victim program’s structure. We believe the matrix organization does not influence the model structure, since overall, the side channel traces are very sparse. In the past few days, we also launched experiences to add noise on the trace or round shift the trace, results are still highly promising. **Please refer to our response to `Is it robust to noise or shifting?` to Reviewer #2.**
>
> **What if there are random zero paddings or other errors in the log?**
>
> Thank you for the question. We have given responses and launched experiences regarding this matter. **Please refer to our response to `Is it robust to noise or shifting?` to Reviewer #2.**
>
> **possible solution and ethics issues**
>
> Thank you very much for the question and suggestion. For our scenario, side channel attacks could be mitigated through software-level or system-level methods. For instance, the victim software might be rewritten into an “input-independent” memory access pattern, thus side channel traces cannot be used to recover any meaningful inputs. However, it is typically not adopted given the extra performance slowdown it can incur.
>
> In addition, our work can also be used as a bug detector to help developers detect and isolate software code blocks and functions that have the risk of private leakage, not merely attacking the target software (**please refer to our response to `what information in the side channel is used for image reconstruction` to Review #2**).
>
> Overall, we will extend the paper in the revision and add discussions on the possible mitigation of our attacks. We will definitely take ethics issues into consideration in the revision. We will upload our revised paper in a few days.

---

### Official Review · AnonReviewer2 · 2020-10-28
**Nice work. But the necessity of using generative models needs further investigation.**

**Rating:** 7
**Confidence:** 3

**Review:**

This paper describes a  new side channel analysis attack framework to reconstruct images from system side channel. This new framework consists of two submodels: the VAE-LP model generates a coarse reconstruction image and the GAN model refines this reconstruction. The experiments show good results in reconstruction.

The idea of reconstructing images from system side channels is interesting and the proposed approach shows good performance. I have the following concerns about the publication of this paper:

1. The novelty is limited from the perspective of the method. The VAE with learned prior is new but little discussion is provided that why a learned prior is necessary. Can we just use a simple supervised end-to-end autoencoder to learn the mapping from side channel signals to the image? Can we use a standard VAE without the Image Encoder branch? Btw, these two are the important baseline for comparison. The authors need to explain why such a design is necessary, or better. Is this a framework specifically designed for SCA, or is it suitable for any task of image reconstruction from any signals?

2. The motivation of the GAN submodel is confusing. As shown in Figure.3, the results of VAE-LP are blurry and are difficult to identify the details, which is reasonable as the input may include very limited information. The GAN submodel adds a lot of details to these images. Some of them are far from the reference images, which is also reasonable as GAN tends to add **texture-like noises**. The refinement or the rich structures and textures are made-up by GAN for good **visual effects**. This additional information is not guaranteed to be correct. Usually, GAN is applied where we require good visual effects. Do we need this in SCA, even more important than the accuracy of the results?

3. The GAN submodel also faces the challenge of **model collapse**, which refers to the problem that the network tends to generate limited patterns to fool the discriminator. As the training dataset is highly structured and the patterns are relatively simple. Is the proposed method capable of working in the scenario of reconstructing natural images? Is the risk of model collapse a concern for the SCA task? This discussion deserves to be included in the paper.

4. Another question that may attract the attention of the machine learning field is, can we know what information in the side channel is used for image reconstruction? In the paper, the side channel information is folded into 2D matrix because of the spatial dependency. Can we visualize this dependency? Is it robust to noise or shifting?

--- Updates ---

After reading the authors' response, I decide to raise my rating to "accept".

---

> ### Author Response · Authors · 2020-11-15
> **Response to Reviewer #2**
>
> Thank you for your efforts reviewing our paper and providing helpful comments and suggestions. We hope you don't mind our long response, and looking forward to hearing your response.
>
> **The novelty is limited from the perspective of the method.**
>
> Thank you for the comments. We wish to clarify the novelty of this work from the following aspects:
>
> - From the perspective of the research idea, this present research, for the first time, apply generative models to infer user private images. Note that conventional secret reconstruction is primarily an “one-to-one” mapping procedure, while we, for the first time, explore the *synthesis* capability of generative models in this field. From a holistic perspective, secrets are reconstructed by **mapping** & **complement**.
> - From the method perspective, we addressed a number of domain-specific challenges and delivered a practical framework to exploit real-world software. In particular, to smoothly learn from side channel traces which are sparse (only a small portion is private-related), lengthy (each trace has over 700K data points), and have locality, we employ a CNN layer. The side channel traces can thus be integrated into the generative framework.
> - From the empirical perspective, our method has already recovered images with highly promising quantitative and qualitative results. We discussed the potential tradeoff of the current design (generalizability). Our work, with code already released, serves as the **baseline** for further research on model structure improvement.
>
> **Two important baseline for comparison.**
>
> Thank you for pointing that out. From the past few days, we launched extra experiences to demonstrate the necessity of *image encoder* and *a learned prior*. We present the synthesized images in the updated Appendix (*Page 26*) of our paper.
>
> Regarding the newly added images on Page 26, we wish to further clarify the following aspects:
>
> 1. Why an image encoder?
> Images synthesized without using the image encoder are seen as low quality --- this is reasonable because the input is merely the matrix derived from the side channel traces.
>
> 2. Why a learned prior?
> When feeding the decoder with a fixed gaussian prior, the synthesized images are seen as low quality as well. Overall, the outputs become not recognizable since the fixed prior has no information of side channel traces. This also indicates that our model is not a simple image generator, and the trace encoder plays a vital role in the pipeline.
>
> 3. Why both?
> As we mentioned in the paper, by incorporating the side channel trace encoder, we can extract latent representations from the logged side channel signals. Simultaneously, by integrating corresponding reference images during training, we provide a guideline to help the image decoder to generate quality images.
>
> We will extend the paper to clarify this design decision and avoid confusions. Kindly advise us if further empirical results regarding this matter is needed. Thank you very much.
>
> **Is this a framework specifically designed for SCA, or is it suitable for any task of image reconstruction from any signals?**
>
> Thank you for the question. The framework is specifically designed for SCA. As aforementioned, the side channel trace is sparse, lengthy, and has locality. We particularly design a CNN layer to handle such traces.

---

> > ### Author Response · Authors · 2020-11-15
> > **Cont'd**
> >
> >
> > **The motivation of the GAN submodel**
> >
> > Thank you for the comments and question. Overall, we decide to use the GAN module via empirical observation. We want to clarify that the design of GAN is from the perspective of “human adversaries.” VAE-LP already shows highly promising results in the quantitative evaluation. GAN’s output could help *qualitative evaluation* by adding visual effects. GAN is indeed not guaranteed to be correct, so we extend the standard GAN loss to force it retain the holistic visual appearance of output from VAE-LP. Our quantitative results (also a fine-grained evaluation of the attributes of faces) show that the accuracy is not affected.
> >
> > Indeed, our evaluation shows that the GAN model helps to exploit user privacy in both quantitative and qualitative SCA analysis:
> >
> > 1. *qualitative*:
> >
> > As discussed in the last paragraph of Section 5.1 Qualitative Evaluation Results (page 6), the GAN module indeed enhances the image quality by adding details and sharpening the blurred regions generated by VAE-LP. In particular, consider the first human gesture case in KTH, where VAE-LP only reconstructs an “black bar” image which does not reveal the human gesture explicitly (although the pixel-level information is indeed within the image). The GAN module properly enhances this obscure image and reconstructs the human gesture, thus violating user privacy.
> >
> > 2. *quantitative*:
> >
> > We gave quantitative comparison in terms of VAE-LP vs. VAE-LP + GAN in Table 8 and Tabel 9 (Appendix; page 18). As reported in the associated text, equipping the GAN module indeed retrain or enhance some quantitative evaluation results:
> >
> > - KTH human gesture dataset: average accuracy changes from 41.9% to 49.8%
> > - CelebA human face dataset: average accuracy changes from 41.0% to 41.6%
> > - LSUN bedroom scene dataset:  average accuracy changes from 12.6% to 12.9%
> >
> > Overall, we will extend the paper and clarify the effectiveness of GAN on the KTH dataset and its marginal improvement on the other two datasets. We will make sure to avoid overclaim or misleadings.
> >
> > Also, we wish to point out a suggestion from Reviewer #3 for using LPIPS, an image-level perceptual loss, to calculate the similarity of the reconstructed image and ground truth image. Results show that the output of VAE-LP & GAN module is much higher than the output of VAE-LP (please see the first table of our response to Review #3). This also indicates that the GAN module is necessary from the perspective of “human adversaries.”
> >
> > **GAN model collapse**
> >
> > Thank you for pointing this out. This work aims to reconstruct the image and exploit user privacy --- the discriminability of generated outputs are our primary focus. We therefore evaluate the accuracy of matching the reconstructed image with the ground truth image rather than the model collapse (we also evaluate the fine-grained attributes of face images). We believe currently the result has manifested that the model collapse is *not* a primary concern of our work.

---

> > > ### Author Response · Authors · 2020-11-15
> > > **Cont'd**
> > >
> > >
> > > **what information in the side channel is used for image reconstruction**
> > >
> > > Thank you for the suggestion! For the past few days we have launched experiences regarding this matter. We explore which data points on the side channel trace affects most to the output by calculating the gradient. Due to the limited time, we tested cache side channel traces logged for the `libjpeg`  software.
> > >
> > > Overall, we are very glad to report that this evaluation helps to pinpoint critical functions in the `libjpeg` software that has input-dependent cache accesses. We are excited to note that this evaluation helps to “re-discover” some functions that have been reported by previous research (mostly with manual effort) as vulnerable to SCA: e.g., the `write_ppm` and `output_ppm` which outputs the decompressed raw image to the disk.
> > >
> > > More importantly, this evaluation helps to pinpoint **NEW** functions that contribute to the private image reconstruction (and hence become vulnerabilities to SCA): `jsimd_can_fdct_islow`, `jsimd_can_fdct_ifast`, `jsimd_convsamp` and `jsimd_convsamp_float`.
> > >
> > > We view this as a highly encouraging finding, in particular:
> > > - As reviewed on Page 3 of our paper, previous research uses manual effort or formal methods to pinpoint program components that depend on inputs, which are program-specific and error-prone with low scalability.
> > > - This research actually reveals a procedure where we leverage gradient to directly highlight which part of the logged side channel trace contributes to the synthesis of outputs. Then, we map the highlighted trace back to where they are derived from in the victim software to isolate vulnerable components (i.e., a “bug detection” tool).
> > >
> > > Overall, the second approach is general and can be launched fully automatically. Again, it not only automatically re-discovers vulnerabilities that were found by previous research (with manual efforts), but helps to identify, to our best knowledge, unknown program components that are vulnerable to SCA.
> > >
> > > We will extend the paper with these experiences (in a few days) to make this research more complete. Looking ahead, we would like to explore this direction as a follow-up of the present work.
> > >
> > > **Is it robust to noise or shifting?**
> > >
> > > Thank you very much for your comments. For the past few days, we have launched experiences to check the robustness to noise or shifting. We explored three settings to 1) add Gaussian noise, 2) replace some randomly selected side channel data points with zero, and 3) round shifting the side channel trace.
> > >
> > > We conduct both quantitative and qualitative analysis. We present the quantitative evaluation results below. **For qualitative results and clarification of experiments details, please refer to the updated Appendix (Page 24-25) of our paper.**
> > >
> > > add Gaussian noise
> > >
> > > | | x=0 | x=0.1 | x=0.2 | x=0.5 |
> > > | :--: | :--: | :--: | :--: | :--: |
> > > | K=1,N=100  | 49.98% | 48.32% | 39.14% | 5.66%  |
> > > | K=5,N=100  | 78.00% | 76.28% | 67.08% | 19.62% |
> > > | K=20,N=100 | 91.98% | 90.56% | 86.22% | 45.20% |
> > >
> > >
> > > Replace some randomly selected side channel data points with zero
> > >
> > > | | x=0 | x=10 | x=20 | x=50 |
> > > | :--: | :--: | :--: | :--: | :--: |
> > > | K=1,N=100  | 49.98% |45.02% | 38.50% | 14.40%  |
> > > | K=5,N=100  | 78.00% |73.30% | 65.52% | 35.50% |
> > > | K=20,N=100 | 91.98% |89.42% | 85.86% | 64.14% |
> > >
> > >
> > > Round shifting the side channel trace
> > >
> > > | | x=0 | x=1 | x=10 | x=100 |
> > > | :--: | :--: | :--: | :--: | :--: |
> > > | K=1,N=100  | 49.98% | 49.56% | 49.94% | 47.48% |
> > > | K=5,N=100  | 78.00% | 77.42% | 77.00% | 75.30% |
> > > | K=20,N=100 | 91.98% | 91.44% | 91.52% | 90.66% |
> > >
> > >
> > >
> > > Overall, we interpret that adding noise does **not** primarily affect the SCA accuracy. We will extend the paper in a few days and add these experiences in the revisions.

---

> > ### Comment · AnonReviewer2 · 2020-11-24
> > **Happy with the results you have achieved**
> >
> > I have read your responses carefully. I want to say that most of my concerns have been well resolved. Combined with the newly updated experiments and discussions, I think this paper reports solid research work. Although the topic of this paper may be difficult for the ICLR community to accept, I want to increase my rating to accept.

---

### Author Response · Authors · 2020-11-15
**Paper Revision**

Dear reviewers, we would like to thank you for all your comments and suggestions.

We have updated our paper with a revision to extend the Appendix. We summarize the main changes below:

1) [Appendix H] Additional evaluations on whether our framework is robust to noise.

2) [Appendix I] Ablation study that investigates the necessity of image encoder and a learned prior.

We will update this paper again in a few days with further revisions (e.g., discuss potential mitigation of our attack, ethics statement), and new experiments (e.g., isolating functions that indeed contribute to the image synthesis).

---

> ### Author Response · Authors · 2020-11-21
> **Paper Revision (2020-11-21)**
>
> Dear reviewers, thank you again for all your comments and suggestions. We have updated our paper. We summarize the *main changes* below:
>
> [3.1 Side Channel Trace Encoding] Clarify the property of side channel traces (sparse, locality, and lengthy) and reason to use CNN.
>
> [3.2 The GAN Model] Clarify the reason to use L$_1$ loss.
>
> [4 Attack Setup] Clarify the side channel trace normalization and explain that typical side channel data points are within a small range.
>
> [4 Attack Setup] Clarify that using a "mega-side channel trace" is generally not aligned with with assumptions of real-world attacks.
>
> [7 Related Work] Add a paragraph to review literatures on SCA mitigation.
>
> [9 Ethics Statement] We added an ethics statement.
>
> [Appendix H] Additional quantitative and qualitative evaluations on whether our framework is robust to noise.
>
> [Appendix I] Ablation study that investigates the necessity of image encoder and a learned prior.
>
> [Appendix I] Ablation study that investigates using image level metrics -- LPIPS.
>
> [Appendix J] New experiments. We use gradient to isolate certain data points on a side channel trace which primarily influence image reconstruction. We further map those "informative" data points back to several functions in `libjpeg`.  We note that this evaluation helps to “re-discover” some functions that have been reported by previous research (mostly with manual effort) as vulnerable to SCA. More importantly, this evaluation helps to pinpoint *unknown* functions that contribute to the private image reconstruction. We view this as a highly interesting and inspiring finding of this research.
>
> **We hope that we have addressed all of your concerns in a satisfactory manner. We sincerely wish to hear from your feedback.**

---

### Decision · Program_Chairs · 2021-01-07
**Final Decision**

**Decision:**

Accept (Poster)

**Comment:**

The paper shows that it is possible to reconstruct private images from CPU cache line and OS page table accesses side channels, using a generative model on top of side channel traces. The reviewers agree that the problem is interesting and the experimental evaluation makes a convincing case that such an attack is possible. The author rebuttal was useful in clarifying some aspects of the paper, and the discussion on possible mitigation strategies is a nice addition to the paper.